# Efficacy in *Galleria mellonella* Larvae and Application Potential Assessment of a New Bacteriophage BUCT700 Extensively Lyse *Stenotrophomonas maltophilia*

Yahao Li,[a] Mingfang Pu,[b] Pengjun Han,[b] Mengzhe Li,[b] Xiaoping An,[b] Lihua Song,[b] ⓘHuahao Fan,[b] Zeliang Chen,[c,d] Yigang Tong[a,b]

[a]Beijing Advanced Innovation Center for Soft Matter Science and Engineering (BAIC-SM), Beijing University of Chemical Technology, Beijing, China
[b]College of Life Science and Technology, Beijing University of Chemical Technology, Beijing, China
[c]Key Laboratory of Tropical Diseases Control, Sun Yat-Sen University, Ministry of Education, Guangzhou, China
[d]Key Laboratory of Zoonose Prevention and Control at Universities of Inner Mongolia Autonomous Region, Tongliao, China

**ABSTRACT** In recent years, *Stenotrophomonas maltophilia* (*S. maltophilia*) has become an important pathogen of clinically acquired infections accompanied by high pathogenicity and high mortality. Moreover, infections caused by multidrug-resistant *S. maltophilia* have emerged as a serious challenge in clinical practice. Bacteriophages are considered a promising alternative for the treatment of *S. maltophilia* infections due to their unique antibacterial mechanism and superior bactericidal ability compared with traditional antibiotic agents. Here, we reported a new phage BUCT700 that has a double-stranded DNA genome of 43,214 bp with 70% GC content. A total of 55 ORFs and no virulence or antimicrobial resistance genes were annotated in the genome of phage BUCT700. Phage BUCT700 has a broad host range (28/43) and can lyse multiple ST types of clinical *S. maltophilia* (21/33). Furthermore, bacteriophage BUCT700 used the Type IV fimbrial biogenesis protein PilX as an adsorption receptor. In the stability test, phage BUCT700 showed excellent thermal stability (4 to 60℃) and pH tolerance (pH = 4 to 12). Moreover, phage BUCT700 was able to maintain a high titer during long-term storage. The adsorption curve and one-step growth curve showed that phage BUCT700 could rapidly adsorb to the surface of *S. maltophilia* and produce a significant number of phage virions. *In vivo*, BUCT700 significantly increased the survival rate of *S. maltophilia*-infected *Galleria mellonella* (*G. mellonella*) larvae from 0% to 100% within 72 h, especially in the prophylactic model. In conclusion, these findings indicate that phage BUCT700 has promising potential for clinical application either as a prophylactic or therapeutic agent.

**IMPORTANCE** The risk of *Stenotrophomonas maltophilia* infections mediated by the medical devices is exacerbated with an increase in the number of ICU patients during the Corona Virus Disease 2019 (COVID-19) epidemic. Complications caused by *S. maltophilia* infections could complicate the state of an illness, greatly extending the length of hospitalization and increasing the financial burden. Phage therapy might be a potential and promising alternative for clinical treatment of multidrug-resistant bacterial infections. Here, we investigated the protective effects of phage BUCT700 as prophylactic and therapeutic agents in *Galleria mellonella* models of infection, respectively. This study demonstrates that phage therapy can provide protection in targeting *S. maltophilia*-related infection, especially as prophylaxis.

**KEYWORDS** bacteriophage (phage) therapy, phage BUCT700, *Stenotrophomonas maltophilia*, phylogenetic analysis, *Galleria mellonella*

Address correspondence to Huahao Fan, fanhuahao@mail.buct.edu.cn, Zeliang Chen, chzl@syau.edu.cn, or Yigang Tong, tong.yigang@gmail.com.

The authors declare no conflict of interest.

**S**tenotrophomonas maltophilia (*S. maltophilia*), an aerobic, motile, nonfermenting, multidrug-resistant, Gram-negative, rod-shaped bacterium, is widely distributed in the natural environment and hospital settings (1). In the last few decades, *S. maltophilia* has developed

into a clinically important opportunistic pathogen (2–4). *S. maltophilia* caused not only disease in clinically severely debilitated or immunocompromised patients but also community-acquired infections in healthy people (5, 6).

*S. maltophilia* is considered to be a rapidly emerging pathogen of concern in hospitals and one of the challenging pathogens in the infectious disease community and studies by the World Health Organization (WHO) (7). *S. maltophilia*-associated clinical symptoms mainly include bacteremia, pneumonia, respiratory tract infections, urinary tract infections, endocarditis, meningitis, eye infections, soft tissue infections, or central venous catheter-related infections (6, 8–11). According to a previous report, infections with *S. maltophilia* were associated with a high rate of mortality, ranging from 21% to 69% (12). Moreover, the ability of *S. maltophilia* to firmly colonize respiratory epithelial cells, medical devices, and therapeutic equipment, considerably increases the risk of infection in hospitalized patients (13, 14). The majority of patients infected with *S. maltophilia* in the intensive care unit (ICU) used medical devices such as mechanical ventilators (82%), intravascular devices (62%), and catheters (86%), as reported by Siripen Kanchanasuwan et al. (15). The risk of *S. maltophilia* infections mediated by the medical devices is exacerbated with an increase in the number of ICU patients during the Corona Virus Disease 2019 (COVID-19) epidemic (13). There have been 12 cases of coinfection with *S. maltophilia* and COVID-19 reported in the research (16). In addition, the most essential factor contributing to the serious infections and high mortality of *S. maltophilia* is its intrinsic resistance to clinically used antibiotics such as penicillins, cephalosporins, carbapenems, aminoglycosides, and macrolides (6). Currently, only a few antibiotics like Sulfamethoxazole-Trimethoprim (SXT) and Levofloxacin can be used for clinical treatment of *S. maltophilia* infection. With extensive use of antibiotics, especially in the hospital environment, *S. maltophilia* rapidly developed resistance to sensitive antibiotics through the acquisition of resistance genes or plasmids and its encoded multidrug efflux pumps (17–19). SXT- and Levofloxacin-resistant *S. maltophilia* are becoming a global epidemic, and the *S. maltophilia* that was resistant to any of the available antibiotics has been isolated by Kamuzu Central Hospital in 2017 (20, 21). Clinical prevention and treatment of multidrug-resistant *S. maltophilia* infections have become a challenging global problem, and it is urgent to find new antimicrobial strategies.

Phage therapy is catching the attention of people as a promising alternative therapy against multidrug-resistant bacterial infections (22). As bacterial viruses, phages feature exquisite host specificity and do not disrupt microbiota *in vivo* as well as lack major end-organ damage. In addition, phages can also replicate and produce themselves within the host bacteria, which prolongs the effectiveness of phage agents (23). However, most of the existing studies about *S. maltophilia* phages rested on genomics and their physiological characteristics (24, 25). Current studies are deficient in systematic assessment of the clinical therapeutic potential of *S. maltophilia* phages and lack data on treatment effects *in vivo* through animal models of infection (26).

Therefore, this study aimed to evaluate the clinical application potential and efficacy of the new *S. maltophilia* bacteriophage BUCT700 *in vitro* and *in vivo*. The genomic and physiological characteristics of the phage BUCT700 were analyzed to examine the potential for clinical utilization. In addition, the *Galleria mellonella* (*G. mellonella*) larvae model was used to investigate the therapeutic effect of BUCT700 *in vivo*, which provides a useful reference for future clinical treatment.

## RESULTS

**Morphological characteristics and host range analysis of BUCT700.** Bacteriophage BUCT700 formed clear plaques with typical virulent phage morphology on its primary host *S. maltophilia* S21 bacterial lawn (Fig. 1A). The results of transmission electron microscopy (TEM) showed that BUCT700 has an icosahedral head with short tails (Fig. 1B). The head diameter was 57.65 $\pm$ 2.04 nm and the length of tails was 9.77 $\pm$ 2.31 nm. Based on the morphology and the latest criteria of International Committee on Taxonomy of Viruses, BUCT700 belonged to the order *Caudovirales* and the unclassified *Okabevirinae* family.

Phage BUCT700 lysed 65.12% of the tested *S. maltophilia* isolated. The lysed bacteria

(A) 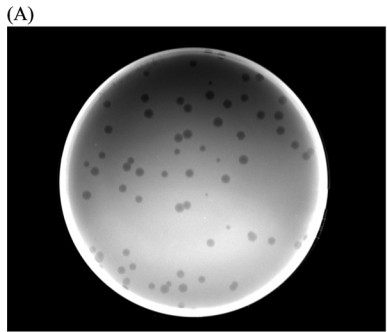     (B) 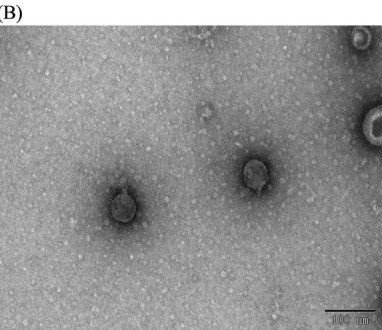

**FIG 1** Morphological characteristics of phage BUCT700. (A) Plaques morphology of phage BUCT700 on *S. maltophilia* S21 bacterial lawn. Bacteriophage BUCT700 formed clear plaques with typical virulent phage morphology on its primary host *S. maltophilia* S21 bacterial lawn; (B) transmission electron micrograph image of phage BUCT700 showed that BUCT700 has an icosahedral head with short tails.

have different sequence types (ST), and the efficiency of plating (EOP) varied from 0.019 to 15.67. The EOP was highest on *S. maltophilia* 532. The detailed information of 43 *S. maltophilia* strains were shown in Table 1.

**Multilocus sequence typing (MLST) analysis of *S. maltophilia* strains.** The sequence type (ST) of *S. maltophilia* isolates was determined by comparing seven pairs of housekeeping genes against the MLST website. According to the analysis of the MLST Database, these 43 strains were classified into thirty-three different ST types, and details were shown in Table 1.

**Replication kinetics and activity of BUCT700.** The results of the experiment about the optimal multiplicity of infection (MOI) showed that the titer of phage BUCT700 at the MOI of 0.0001 was higher than other MOIs, indicating that the MOI of 0.0001 was most suitable for the growth of phage BUCT700 (Fig. 2A).

The absorption rate of phage BUCT700 onto *S. maltophilia* S21 was investigated, and the result was shown in Fig. 2B. About 80.40% of the phage particles were adsorbed after 2 min, with more than 95% being absorbed after 14 min. Then, the phage started to release after 18 min. Compared with *S. maltophilia* phage Smp14 and phiSM5, phage BUCT700 has faster adsorption and higher adsorption rate (24, 27).

The infection kinetics of phage BUCT700 were determined by a one-step growth curve. The latent and lytic periods of BUCT700 were approximately 10 min and 40 min, respectively (Fig. 2C). The latent and burst period of BUCT700 are shorter compared with *S. maltophilia* phage BUCT555 (28).

**Stability of phage BUCT700.** In order to determine the potential of making phage BUCT700 into formulations for clinical use, thermal, pH, and long-term stability of phage BUCT700 were tested. The phage BUCT700 showed high stability at different temperatures for the first 6 h. At 4, 20, 30, 40, and 50°C, phage titer remained stable for up to 24 h. But the titer of phage decreased significantly at 60 and 70°C after 12 h, and it is almost inactive at 70°C for 24 h (Fig. 3A). Phage BUCT700 had a high activity with pH 4 to 12 until 24 h, but the phage titer decreased rapidly starting from 6 h after incubation at pH = 2 (Fig. 3B). Additionally, the long-term stability of phage BUCT700 at 4°C was monitored monthly for 5 months. The results showed no significant reduction of phage titer over 5 months, indicating good long-term stability of phage BUCT700 (Fig. 3C).

**Adsorption receptor identification.** To identify potential receptors for bacteriophage BUCT700 infection, we isolated phage-resistant mutant S21R and analyzed the mutations from next-generation sequencing data of phage-resistant mutant S21R by using *breseq*. The results of *breseq* showed that one base mutation in the PilX gene (AGC → AGA) compared with S21, resulted in a mutation from arginine to serine. To verify the results of mutation analysis, we constructed a pBBR1MCS-2-PilX plasmid and electric transferred it into phage-resistant mutant S21R to obtain the complementary strain S21R + PilX. The results of plaque and adsorption assays on phage-resistant mutant S21R and complementary strain S21R + PilX showed that BUCT700 no longer adsorbed and lysed S21R, whereas the complementary

**TABLE 1** Detail information of host range of phage BUCT700[a]

| Species | Strain | Origin | ST type | Sensitivity | EOP |
|---|---|---|---|---|---|
| *S. maltophilia* | 34 | 307 hospital | ST 418 | + | 0.683 |
| *S. maltophilia* | 35 | 307 hospital | ST 461 | + | 0.833 |
| *S. maltophilia* | 118 | 307 hospital | ST 4 | + | 1.083 |
| *S. maltophilia* | 209 | 307 hospital | ST 463 | − | − |
| *S. maltophilia* | 532 | 307 hospital | ST 296 | + | 15.67 |
| *S. maltophilia* | 548 | 307 hospital | ST 190 | + | 1.45 |
| *S. maltophilia* | 690 | 307 hospital | ST 115 | + | 0.134 |
| *S. maltophilia* | 824 | 307 hospital | ST 413 | − | − |
| *S. maltophilia* | 826 | 307 hospital | ST 378 | − | − |
| *S. maltophilia* | 992 | 307 hospital | ST 8 | + | 5.2 |
| *S. maltophilia* | 1207 | 307 hospital | ST 502 | + | 0.317 |
| *S. maltophilia* | 1209 | 307 hospital | ST 7 | − | − |
| *S. maltophilia* | 1284 | 307 hospital | ST 138 | − | − |
| *S. maltophilia* | 1785 | 307 hospital | ST 31 | + | 0.02 |
| *S. maltophilia* | 1786 | 307 hospital | ST 362 | + | 10 |
| *S. maltophilia* | S1 | Aviation General Hospital | ST 564 | + | 0.062 |
| *S. maltophilia* | S2 | Aviation General Hospital | ST 394 | + | 0.976 |
| *S. maltophilia* | S3 | Aviation General Hospital | ST 634 | + | 1.35 |
| *S. maltophilia* | S4 | Aviation General Hospital | ST 503 | + | 1.6 |
| *S. maltophilia* | S5 | Aviation General Hospital | ST 634 | + | 1.233 |
| *S. maltophilia* | S6 | Aviation General Hospital | ST 631 | − | − |
| *S. maltophilia* | S7 | Aviation General Hospital | ST 138 | − | − |
| *S. maltophilia* | S8 | Aviation General Hospital | ST 133 | + | 1.7 |
| *S. maltophilia* | S9 | Aviation General Hospital | ST 413 | − | − |
| *S. maltophilia* | S10 | Aviation General Hospital | ST 634 | + | 1.117 |
| *S. maltophilia* | S11 | Aviation General Hospital | ST 634 | + | 0.7 |
| *S. maltophilia* | S12 | Aviation General Hospital | ST 324 | + | 0.019 |
| *S. maltophilia* | S13 | Aviation General Hospital | ST 634 | + | 1.717 |
| *S. maltophilia* | S14 | Aviation General Hospital | ST 828 | + | 1.25 |
| *S. maltophilia* | S15 | Aviation General Hospital | ST 223 | + | 1.15 |
| *S. maltophilia* | S16 | Aviation General Hospital | ST 634 | + | 1.383 |
| *S. maltophilia* | S17 | Aviation General Hospital | ST 291 | − | − |
| *S.maltophilia* | S18 | Aviation General Hospital | ST 644 | − | − |
| *S. maltophilia* | S19 | Aviation General Hospital | ST 713 | − | − |
| *S. maltophilia* | S20 | Aviation General Hospital | ST 151 | − | − |
| *S. maltophilia* | S21 | Aviation General Hospital | ST 4 | + | 1 |
| *S. maltophilia* | S22 | Aviation General Hospital | ST 281 | + | 0.45 |
| *S. maltophilia* | S23 | Aviation General Hospital | ST 634 | + | 5.167 |
| *S. maltophilia* | S24 | Aviation General Hospital | ST 167 | − | − |
| *S. maltophilia* | S2016 | Aviation General Hospital | ST 139 | − | − |
| *S. maltophilia* | S2030 | Aviation General Hospital | ST 116 | + | 0.082 |
| *S. maltophilia* | S2033 | Aviation General Hospital | ST 139 | − | − |
| *S. maltophilia* | S532 | Aviation General Hospital | ST 50 | + | 1.1 |

[a]"+," susceptible; "−," resistance.

strain S21R + PilX regained the sensitivity to BUCT700, indicating the Type IV fimbrial biogenesis protein PilX is an adsorption receptor of phage BUCT700 (Fig. 4).

**Genomic analysis and annotation of phage BUCT700.** In general, bioinformatics analysis can predict the biological properties and safety of phages used for medical purposes. The genome of phage BUCT700 was sequenced and analyzed by various bioinformatics tools. The full length of linear double-stranded DNA (dsDNA) of phage BUCT700 is 43,214 bp with 70% GC content. The whole genome of phage BUCT700 was deposited in the GenBank database with the accession number OM735686.1. Based on RAST and BLASTp analysis results, the genome contains 55 predicted open reading frames (ORFs). A total of 54 ORFs were presented on the positive strand, with 1 ORF on the negative strand (Fig. 5, Table 2). The majority of the ORFs presented an ATG start codon (90.90%), while 7.30% started with GTG and 1.80% started with TTG. The length of the nucleotide coding sequence ranges from 137 to 3971 bp, which corresponds to the protein sequence length of 45 to 1323 amino acids. In brief, ORFs account for 39,701 bp and the gene density is as high as 91.87%. Of the 55 ORFs predicted, only 16

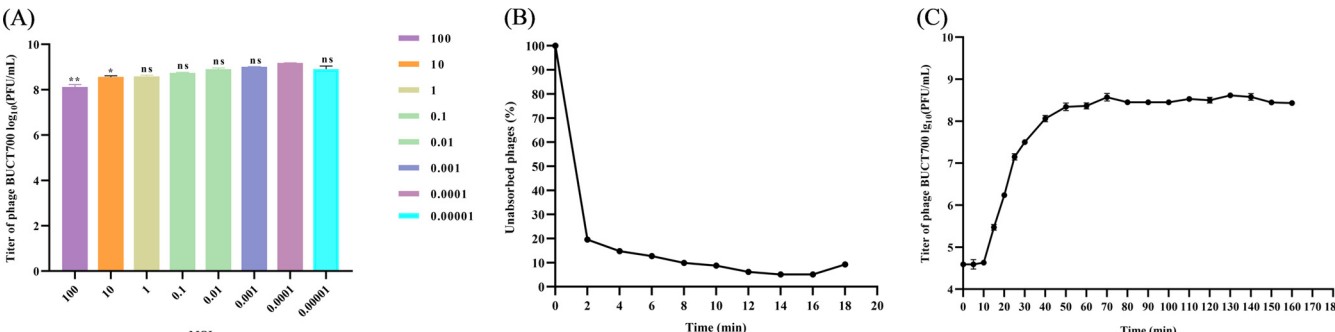

**FIG 2** Physiological characterization of phage BUCT700. (A) Optimal MOI assays of phage BUCT700; (B) the adsorption curve of phage BUCT700; (C) the One-step growth curve of BUCT700. Data are shown as the mean ± SD. **, $P < 0.01$ and *, $P < 0.05$ indicate a significant difference between this group and control (MOI = 0.0001), whereas ns indicates no significant difference.

encode proteins with functions associated with phage lysis, structure, replication, transcription, and packaging. The remaining genes were annotated as hypothetical proteins or not hit. No tRNA, virulence, or drug-resistant genes were searched. The whole-genome sequence alignment of phage BUCT700 showed that phage BUCT700 has a high nucleotide sequence similar to *S. maltophilia* phage BUCT598 (identity: 81.86% and query cover: 79% with E value:0) and *S. maltophilia* phage P15 (identity: 81.76% and query cover: 78% with E value:0). The results of global comparisons to BUCT598 and P15 showed that both of them have a lot of hypothetical proteins and the highly similar regions are mainly hypothetical proteins (Fig. 6).

The genomic annotation of phage BUCT700 showed that 7 ORFs were associated with replication and transcription, including DNA primase/helicase (ORF 20), DNA helicase (ORF 21), DNA polymerase I (ORF 22), 5′–3′ exonuclease (ORF 24), DNA exonuclease (ORF 25), exonuclease (ORF 26), and DNA-dependent RNA polymerase (ORF 34). During DNA replication, a DNA helicase and a DNA polymerase cooperatively unwind the parental DNA (29). Although DNA polymerase I (ORF 22) has exonuclease activity, phage BUCT700 still carries 3 genes (ORF 24, 25, and 26) encoding exonucleases. ORF 24 encodes 5′–3′ exonuclease with a defined direction of action, while ORF 25 and 26 neither give an exact direction of action. ORF 24, 25, and 26 play roles in DNA replication, DNA mismatch repair (MMR), and DNA double-stranded break repair (DSBR) (30). After RNA translation, structural proteins and functional proteins associated with packaging and lysis were expressed. There are 5 genes encoding structural proteins, including major capsid protein (ORF 39), tail tubular protein A (ORF 41), and tail fiber (ORF 46, 47, and 49). It is worth noting that BUCT700 has many genes (ORF 46, 47, and 49) encoded tail fiber that recognizes host receptor proteins during the infestation.

To analyze the phage evolutionary relationship, the phylogenetic tree of conserved proteins was constructed for comparative analysis using Molecular Evolutionary Genetic Analysis (MEGA) v7.0 (31). The sequence of DNA polymerase I, terminase small subunit, and terminase large subunit proteins of phage BUCT700 were all closely related to *S.*

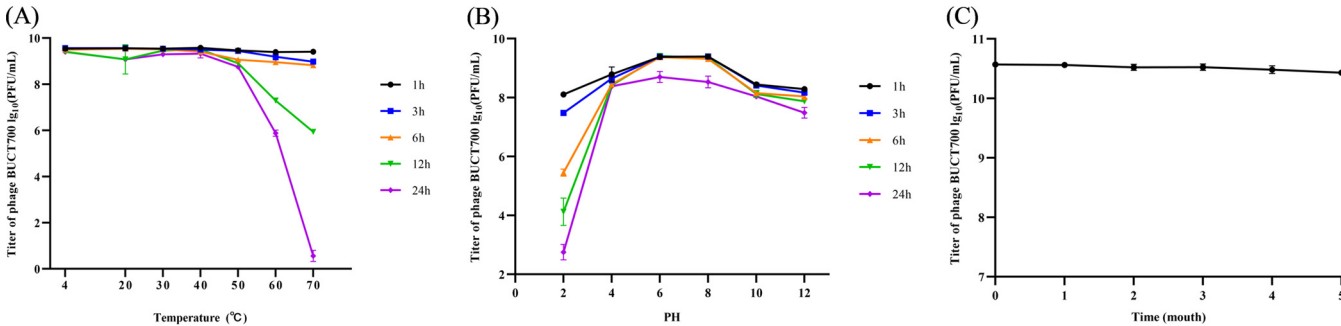

**FIG 3** Stability of phage BUCT700. (A) Temperature stability of phage BUCT700; (B) pH stability of phage BUCT700; (C) long-term stability of phage BUCT700. Results are presented as mean values ± SD.

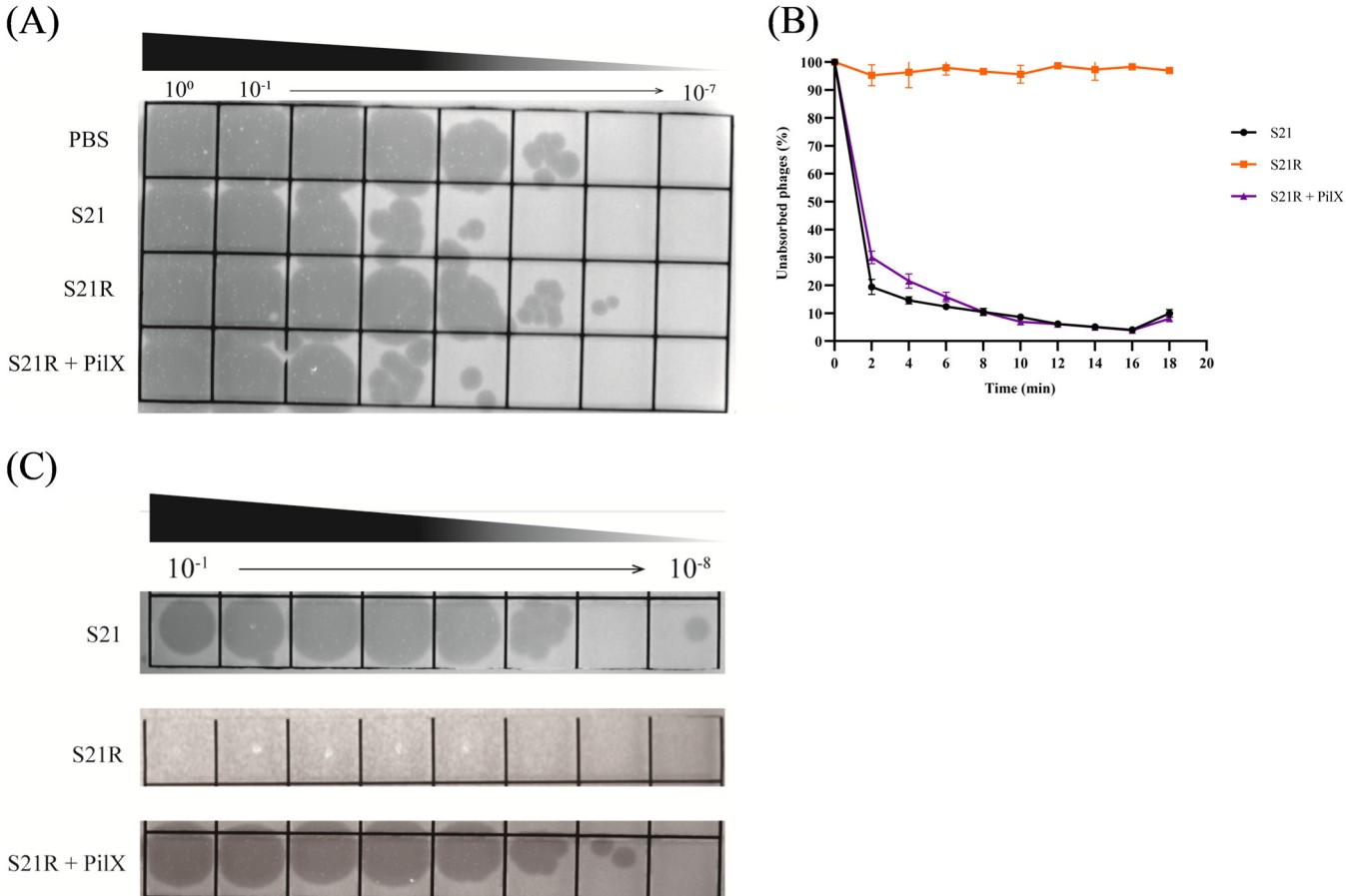

**FIG 4** Adsorption receptor identification of BUCT700. Bacteriophage BUCT700 uses Type IV fimbrial biogenesis protein PilX as an adsorption receptor to infect *S. maltophilia* S21. (A) Compared with *S. maltophilia* S21, phage-resistant mutant S21R produced significantly impaired adsorption, whereas the complementary strain S21R + PilX could be readsorbed by phage BUCT700. (B) Phage BUCT700 can rapidly adsorb *S. maltophilia* S21, whereas BUCT700 cannot adsorb phage-resistant mutant S21R. When complementation of phage-resistant mutant S21R with the PilX gene was performed, phage BUCT700 restores adsorption to the complementary strain S21R + PilX. (C) *S. maltophilia* S21 is susceptible to BUCT700 infection, whereas phage-resistant mutant S21R is resistant to phage infection, but the complementary strain S21R + PilX restores phage infection to *S. maltophilia* S21 levels. Data are shown as the mean ± SD.

*maltophilia* phage BUCT598 and *S. maltophilia* phage P15 (Fig. 7A–C). The mechanism of DNA replication and packaging in phage BUCT700 may be similar to *S. maltophilia* phage BUCT598 and *S. maltophilia* phage P15 (32).

**Assessment of the efficacy of phage BUCT700 against *S. maltophilia* S21 in *vitro* and *vivo*.** The lytic activity of phage BUCT700 with *S. maltophilia* S21 was determined at different MOIs for 14 h with uninfected *S. maltophilia* S21 as a control. The results showed that the absorbance of uninfected control group S21 consistently increased and the absorbance of phage BUCT700 infected S21 at different MOIs was significantly reduced in the first 1 h compared with the control group. And the group of phage BUCT700 infected S21 at MOI of 10, 1, 0.1, 0.01, 0.001, 0.0001 maintained a low absorbance for up to 11 h (Fig. 8A). Subsequently, the absorbance of phage BUCT700 infected S21 at different MOIs increased, indicating that phage-resistant bacteria appeared within 10 h to 11 h.

Based on survival curves of *G. mellonella* after injecting with different concentrations of S21, S21 at a dose of $5 \times 10^8$ CFU/mL caused significant mortality at 24 h. And the *G. mellonella* die too quickly to be observed at a dose of $1 \times 10^9$ CFU/mL. To evaluate the potential efficacy of bacteriophage BUCT700 against *S. maltophilia* in the clinical context, a dose of $5 \times 10^8$ CFU/mL was selected as the infection dose (Fig. 8B). For the treatment experiments, all *G. mellonella* larvae as a positive-control group died within 24 h (Fig. 9A). The group of only injected PBS and only injected phage at an MOI of 100 survived for 72 h, indicating that either PBS or phage did not cause any death of *G. mellonella* larvae (Fig. 9B and 9C). The results of evaluating phage BUCT700 as potential therapeutic agents showed

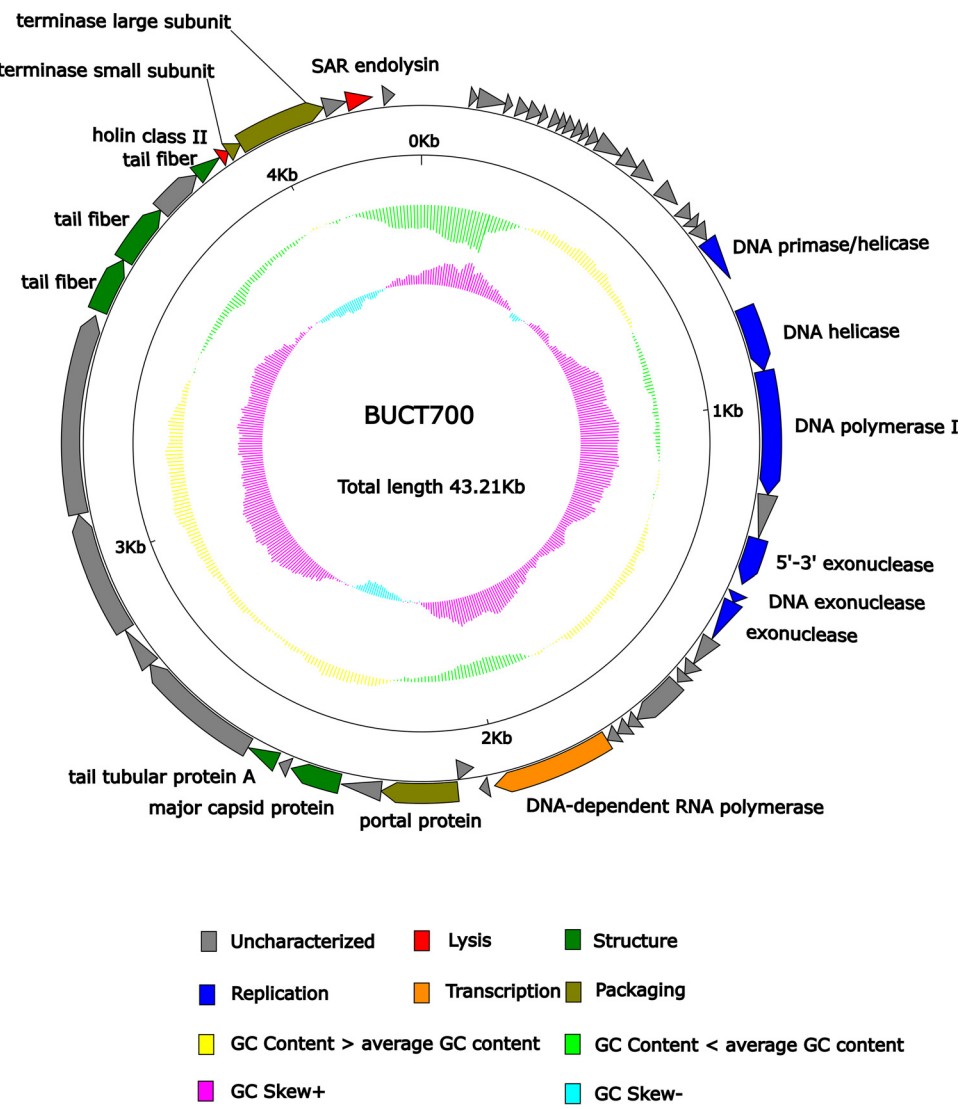

**FIG 5** Schematic diagram of the genome of bacteriophage BUCT700. All predicted ORFs are represented by arrows, where the positive direction is in the outer ring and the negative direction is in the inner ring. Red, lysis; orange, transcription; deeply green, structure; brown, packaging; deeply blue, replication; gray, uncharacterized. Yellow represents greater than the average GC content, and the light green represents the opposite. Purple represents G-C/G+C > 0, light blue represents < 0.

that the survival rate of *G. mellonella* larvae injected phages at MOI of 100, 10, 1, and 0.1 was 100%, 86.67%, 50.00%, and 33.33%, respectively (Fig. 8C, 9D–G). All of the group injected phages were able to protect *G. mellonella* larvae and good therapeutic efficacy was shown in high titer conditions (MOI of 100 and 10). When pretreating with phage with MOI of 100, 10, 1, and 0.1, the survival rate of *G. mellonella* larvae was 100%, 96.15%, 93.33%, and 81.25%, respectively (Fig. 8D, 9H–K). In the prophylactic treatment model of phage, phage BUCT700 provided good protection as a prophylactic agent.

## DISCUSSION

*S. maltophilia* has become a clinically significant opportunistic pathogen and is associated with significant morbidity and mortality. In 2009, the 16 invasive *S. maltophilia* strains isolated from blood cultures were distributed among nine different ST types, including ST4 (33). The results indicated that ST4 type *S. maltophilia* strains can survive and multiply in the blood, causing bacteremia after infection. *S. maltophilia* S21, isolated from the hospital environment and belonging to the ST4 type, has the potential to cause complications of bacteremia in

**TABLE 2** Predicted ORFs in the genome of phage BUCT700

| ORF | Strand | Start | Stop | Length (AA) | Putative function | Best-match BLASTp result | Query cover (%) | E-values | Identity (%) | Accession | MW (kDa) |
|---|---|---|---|---|---|---|---|---|---|---|---|
| ORF1 | + | 959 | 1108 | 49 | Hypothetical protein | Stenotrophomonas phage BUCT609 | 100 | 9e-21 | 83.67 | QVR48685.1 | 5.8 |
| ORF2 | + | 1122 | 1676 | 184 | Hypothetical protein | Stenotrophomonas phage vB_SmaS_P15 | 97 | 5e-118 | 91.67 | UFI08392.1 | 20.3 |
| ORF3 | + | 1669 | 1812 | 47 | Hypothetical protein | Burkholderia multivorans | 97 | 7e-10 | 63.83 | WP_212166571.1 | 5.6 |
| ORF4 | + | 1900 | 2148 | 82 | No hits | | | | | - | 9.0 |
| ORF5 | + | 2145 | 2399 | 84 | No hits | | | | | - | 9.5 |
| ORF6 | + | 2396 | 2539 | 47 | Hypothetical protein | Stenotrophomonas phage BUCT598 | 91 | 1e-14 | 79.07 | QWT56549.1 | 5.3 |
| ORF7 | + | 2615 | 2803 | 62 | Hypothetical protein | Stenotrophomonas phage BUCT598 | 90 | 9e-25 | 87.50 | QWT56550.1 | 7.2 |
| ORF8 | + | 2790 | 2939 | 49 | Hypothetical protein | Stenotrophomonas phage BUCT598 | 100 | 4e-20 | 75.51 | QWT56551.1 | 5.8 |
| ORF9 | + | 2926 | 3120 | 64 | Hypothetical protein | Stenotrophomonas phage vB_SmaS_P15 | 100 | 1e-23 | 67.65 | UFI08398.1 | 7.3 |
| ORF10 | + | 3117 | 3290 | 57 | Hypothetical protein | Stenotrophomonas phage Ptah | 85 | 2e-04 | 51.79 | QYW01724.1 | 6.3 |
| ORF11 | + | 3287 | 3424 | 45 | Hypothetical protein | Stenotrophomonas phage Ptah | 100 | 2e-20 | 86.67 | QYW01725.1 | 5.1 |
| ORF12 | + | 3447 | 3638 | 63 | Hypothetical protein | Stenotrophomonas phage TS-10 | 100 | 2e-20 | 65.08 | UDL16930.1 | 7.1 |
| ORF13 | + | 3642 | 4178 | 178 | Hypothetical protein | Stenotrophomonas phage BUCT609 | 63 | 2e-18 | 45.22 | QVR48675.1 | 20.1 |
| ORF14 | + | 4178 | 4561 | 127 | Hypothetical protein | Stenotrophomonas phage vB_SmaS_P15 | 72 | 2e-34 | 65.22 | UFI08402.1 | 14.6 |
| ORF15 | + | 4561 | 4968 | 135 | Hypothetical protein | Stenotrophomonas phage vB_SmaS_P15 | 78 | 3e-59 | 94.34 | UFI08403.1 | 15.2 |
| ORF16 | + | 5167 | 5634 | 155 | Hypothetical protein | Xylella phage Paz | 89 | 5e-50 | 63.04 | YP_008858885.1 | 15.7 |
| ORF17 | + | 5762 | 6022 | 86 | Hypothetical protein | Stenotrophomonas phage BUCT598 | 100 | 5e-27 | 61.63 | QWT56558.1 | 9.9 |
| ORF18 | + | 6037 | 6219 | 60 | Hypothetical protein | Stenotrophomonas phage vB_SmaS_P15 | 95 | 3e-16 | 61.40 | UFI08405.1 | 6.6 |
| ORF19 | + | 6219 | 6542 | 107 | Hypothetical protein | Stenotrophomonas phage BUCT598 | 100 | 6e-43 | 61.68 | QWT56560.1 | 12.2 |
| ORF20 | + | 6554 | 7438 | 294 | DNA primase/helicase | Stenotrophomonas phage vB_SmaS_P15 | 100 | 7e-172 | 80.00 | UFI08408.1 | 33.0 |
| ORF21 | + | 8060 | 9367 | 435 | DNA helicase | Stenotrophomonas phage vB_SmaS_P15 | 100 | 0.0 | 79.31 | UFI08410.1 | 49.5 |
| ORF22 | + | 9369 | 11819 | 816 | DNA polymerase I | Stenotrophomonas phage vB_SmaS_P15 | 100 | 0.0 | 83.25 | UFI08411.1 | 92.4 |
| ORF23 | + | 11823 | 12695 | 290 | Hypothetical protein | Stenotrophomonas phage BUCT598 | 100 | 9e-145 | 79.45 | QWT56567.1 | 32.2 |
| ORF24 | + | 12706 | 13650 | 314 | 5'-3' exonuclease | Stenotrophomonas phage BUCT598 | 97 | 4e-144 | 66.23 | QWT56568.1 | 35.5 |
| ORF25 | + | 13847 | 14047 | 66 | DNA exonuclease | Stenotrophomonas phage BUCT609 | 96 | 4e-33 | 87.50 | QVR48660.1 | 7.7 |
| ORF26 | + | 14044 | 14880 | 278 | Exonuclease | Stenotrophomonas phage Ponderosa | 98 | 0.0 | 86.91 | QEG09743.1 | 32.3 |
| ORF27 | + | 14890 | 15492 | 200 | Hypothetical protein | Stenotrophomonas phage BUCT598 | 98 | 3e-119 | 86.00 | QWT56571.1 | 21.6 |
| ORF28 | + | 15489 | 15755 | 88 | Hypothetical protein | Stenotrophomonas phage vB_SmaS_P15 | 100 | 1e-51 | 90.91 | UFI08417.1 | 10.0 |
| ORF29 | + | 15748 | 15963 | 71 | Hypothetical protein | Xanthomonas phage phi Xc10 | 95 | 2e-35 | 86.76 | YP_009791547.1 | 8.1 |
| ORF30 | + | 15993 | 17048 | 351 | Hypothetical protein | Stenotrophomonas phage vB_SmaS_P15 | 100 | 0.0 | 87.75 | UFI08418.1 | 39.9 |
| ORF31 | + | 17045 | 17272 | 75 | Hypothetical protein | Stenotrophomonas phage vB_SmaS_P15 | 97 | 7e-40 | 86.30 | UFI08419.1 | 8.8 |
| ORF32 | + | 17269 | 17523 | 84 | Hypothetical protein | Stenotrophomonas phage vB_SmaS_P15 | 97 | 2e-31 | 69.05 | UFI08420.1 | 10.0 |
| ORF33 | + | 17525 | 17758 | 77 | Hypothetical protein | Xanthomonas phage Xaa_vB_phi31 | 98 | 1e-12 | 49.38 | QOI69531.1 | 9.4 |
| ORF34 | + | 17761 | 20160 | 799 | DNA-dependent RNA polymerase | Stenotrophomonas phage BUCT598 | 100 | 0.0 | 81.48 | QWT56577.1 | 91.6 |
| ORF35 | + | 20276 | 20461 | 61 | Hypothetical protein | Stenotrophomonas phage BUCT598 | 96 | 4e-31 | 89.83 | QWT56578.1 | 6.8 |
| ORF36 | − | 20857 | 20510 | 115 | No hits | | | | | - | 12.6 |
| ORF37 | + | 20886 | 22406 | 506 | portal protein | Stenotrophomonas phage BUCT598 | 100 | 0.0 | 88.74 | QWT56580.1 | 56.7 |
| ORF38 | + | 22403 | 23203 | 266 | Hypothetical protein | Stenotrophomonas phage vB_SmaS_P15 | 85 | 9e-145 | 88.60 | UFI08426.1 | 27.7 |
| ORF39 | + | 23231 | 24229 | 332 | Major capsid protein | Stenotrophomonas phage BUCT598 | 100 | 0.0 | 96.39 | QWT56582.1 | 36.2 |
| ORF40 | + | 24274 | 24489 | 71 | Hypothetical protein | Stenotrophomonas phage BUCT609 | 98 | 3e-33 | 83.10 | QVR48648.1 | 7.8 |
| ORF41 | + | 24544 | 25164 | 206 | Tail tubular protein A | Stenotrophomonas phage BUCT598 | 99 | 2e-132 | 87.25 | QWT56584.1 | 23.8 |
| ORF42 | + | 25174 | 27711 | 845 | Hypothetical protein | Stenotrophomonas phage vB_SmaS_P15 | 100 | 0.0 | 83.94 | UFI08430.1 | 93.6 |
| ORF43 | + | 27711 | 28538 | 275 | Internal virion protein | Stenotrophomonas phage BUCT598 | 100 | 2e-176 | 88.04 | QWT56586.1 | 29.9 |
| ORF44 | + | 28549 | 30999 | 816 | Internal virion protein | Stenotrophomonas phage BUCT598 | 100 | 0.0 | 84.93 | QWT56587.1 | 90.0 |

**TABLE 2** (Continued)

| ORF | Strand | Start | Stop | Length (AA) | Putative function | Best-match BLASTp result | Query cover (%) | E-values | Identity (%) | Accession | MW (kDa) |
|---|---|---|---|---|---|---|---|---|---|---|---|
| ORF45 | + | 31011 | 34982 | 1323 | Hypothetical protein | Stenotrophomonas phage vB_SmaS_P15 | 100 | 0.0 | 82.19 | UFI08433.1 | 143.3 |
| ORF46 | + | 35088 | 36206 | 372 | Tail fiber | Stenotrophomonas phage Pepon | 100 | 0.0 | 83.87 | QYW01993.1 | 40.9 |
| ORF47 | + | 36209 | 37432 | 407 | Tail fiber | Stenotrophomonas phage Pepon | 100 | 0.0 | 86.24 | QYW01994.1 | 45.4 |
| ORF48 | + | 37429 | 38418 | 329 | Hypothetical protein | Stenotrophomonas phage Pepon | 100 | 0.0 | 97.26 | QYW01995.1 | 35.8 |
| ORF49 | + | 38415 | 38987 | 190 | Tail fiber | Stenotrophomonas phage Pepon | 100 | 2e-135 | 96.84 | QYW01996.1 | 21.8 |
| ORF50 | + | 38990 | 39199 | 69 | Holin class II | Stenotrophomonas phage BUCT598 | 100 | 3e-29 | 72.46 | QWT56537.1 | 7.7 |
| ORF51 | + | 39177 | 39482 | 101 | Terminase small subunit | Stenotrophomonas phage BUCT598 | 95 | 9e-33 | 64.95 | QWT56538.1 | 10.5 |
| ORF52 | + | 39469 | 41265 | 598 | Terminase large subunit | Stenotrophomonas phage BUCT598 | 99 | 0.0 | 87.21 | QWT56539.1 | 66.5 |
| ORF53 | + | 41268 | 41723 | 151 | Hypothetical protein | Stenotrophomonas phage BUCT609 | 100 | 9e-89 | 95.36 | QVR48635.1 | 16.7 |
| ORF54 | + | 41733 | 42242 | 169 | SAR endolysin | Stenotrophomonas phage BUCT598 | 98 | 2e-88 | 75.90 | QWT56542.1 | 19.2 |
| ORF55 | + | 42458 | 42688 | 76 | Hypothetical protein | Stenotrophomonas phage vB_SmaS_P15 | 92 | 4e-23 | 61.43 | UFI08388.1 | 8.6 |

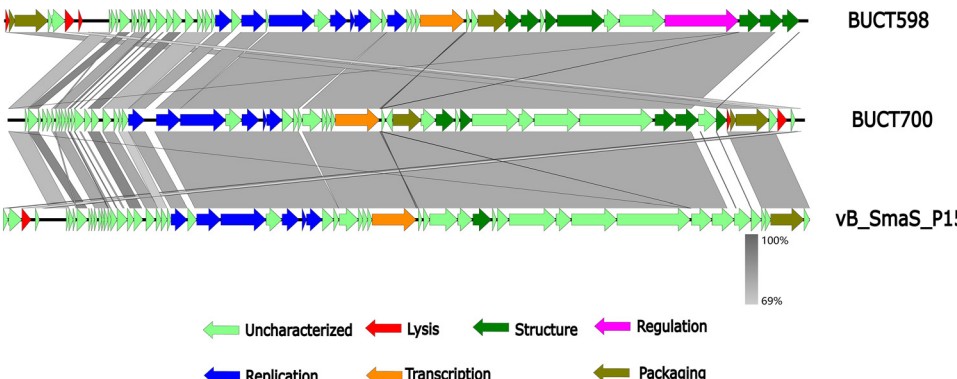

**FIG 6** Multiple-sequence alignment of phage genomes. The whole genome of *S. maltophilia* phage BUCT700, *S. maltophilia* phage BUCT598 and *S. maltophilia* phage P15 were compared using Easyfig. The gray shading indicates sequence similarities between the genomes.

ICU patients. Considering the high mortality rate caused by *S. maltophilia*, S21 requires the development of new antimicrobial agents. This study provides a novel phage BUCT700 and analyzes its clinical application potential *in vitro* and *in vivo*.

Phage BUCT700 formed clearly visible plaques on the lawn of *S. maltophilia* S21 and belongs to the unclassified *Okabevirinae* family indicated by TEM and bioinformatics analysis. Phage BUCT700 could infect 65.12% of *S. maltophilia* isolates, demonstrating that phage BUCT700 has a wide lytic range. And phage BUCT700 can lyse twenty-one ST species of *S. maltophilia* strains, including ST 4, 8, 31, 50, 115, 116, 133, 190, 223, 281, 296, 324, 362, 394, 418, 461, 502, 503, 564, 634, and 828. And most ST types of those *S. maltophilia* strains were reported in clinical patients or hospital environment (14, 33). Further study on the adsorption receptor of phage BUCT700 revealed that BUCT700 can adsorb bacteria using Type IV fimbrial biogenesis protein PilX. PilX highly conserved in bacterial genome was implicated as key promoters of pilus assembly on the cell surface and played a key role in pathogenesis in a variety of bacterial species (34). Based on the research by Victoria A Marko et al., mutation or loss of PilX had reduced virulence and motility of bacteria (35). Furthermore, PilX is a pilus-associated protein essential for bacterial aggregation which confers a powerful biofilm-forming ability on *S. maltophilia* (36). Bacteriophage BUCT700, which uses PilX as an adsorbed receptor, is able to extensively infest clinical *S. maltophilia* and produces survival pressure causing changes to PilX resulting in reduced virulence of *S. maltophilia*. Therefore, we believe that BUCT700 has good potential as clinical antimicrobial agents.

At the genetic level, the phage genome characteristics and safety analysis for clinical use were analyzed by bioinformatic tools. The genome of phage BUCT700 is a 43,214 bp long dsDNA with 70% GC content. The phage BUCT700 has a high nucleotide sequence similarity to *S. maltophilia* phage BUCT598 and *S. maltophilia* phage P15, indicating BUCT700 may have similar life cycles with them. Interestingly, most of the 16 annotated genes of phage BUCT700 are related to DNA replication and DNA packaging. Three genes in the genome of BUCT700 encode exonucleases, significantly more than *S. maltophilia* phage BUCT598 and P15. Exonuclease can repair DNA and ensure rapid and correct DNA synthesis indicating that phage BUCT700 has a precise DNA synthesis system to adapt to the needs of rapid replication. These three exonucleases (ORF 24, 25, and 26) of phage BUCT700 make it difficult to carry external virulence and drug resistance genes, improving the safety of its clinical use. In addition, no virulence or drug-resistance genes were searched in the complete genome of phage BUCT700. It is worth noting that BUCT700 has many genes (ORF 46, 47, and 49) encoded tail fiber which recognizes host receptor proteins during the infestation. More tail fiber proteins may provide a variety the host receptors, which may contribute to its rapid adsorption and broad host spectrum. The genomic analysis of phage BUCT700 showed that it has no disadvantages for clinical use.

The main influences on the clinical application of phages are external environmental factors. Considering the clinical application environment and the need for making therapeutic drugs,

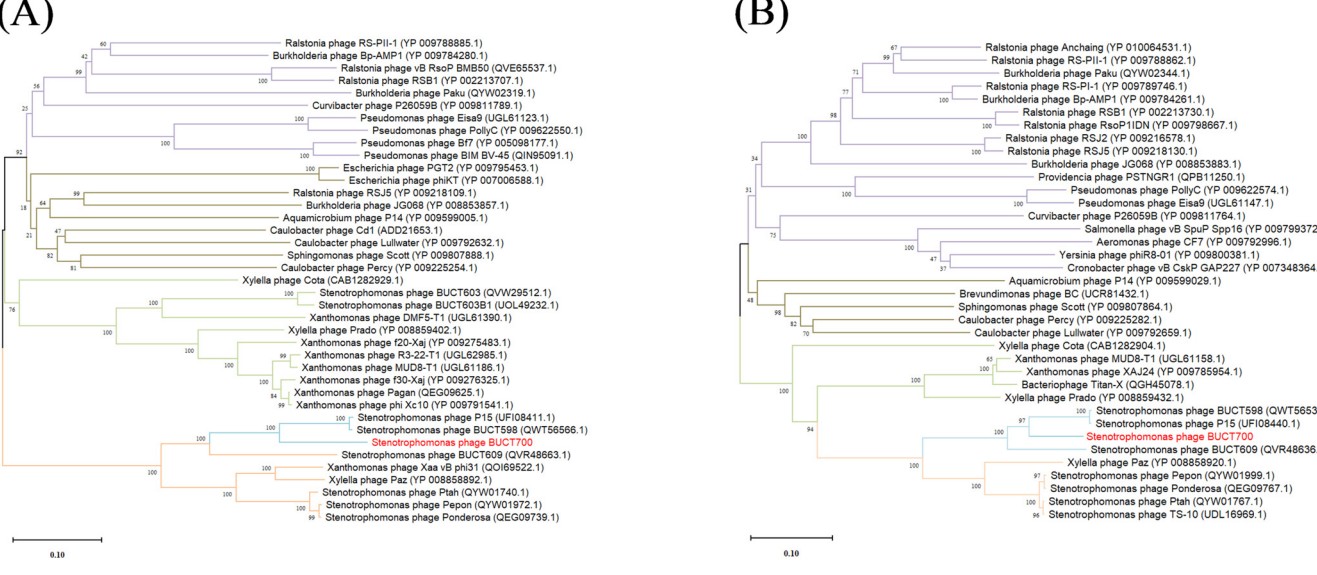

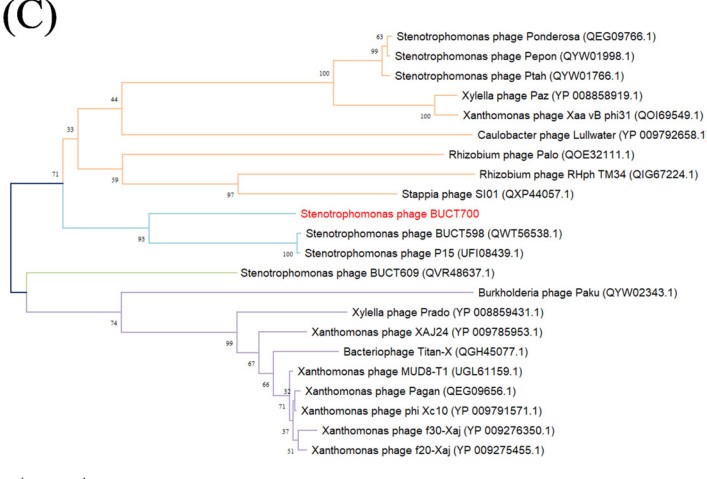

**FIG 7** Phylogenetic relationship between selected phage amino acid sequences. (A) Tree assembled with DNA polymerase sequences; (B) Tree assembled with terminase large subunit sequence; (C) Tree assembled with terminase small subunit sequence. The *S. maltophilia* phage is marked in red. The phylogenetic tree was checked by Bootstrap method, and the number of test replicates is 1,000 times.

temperature and pH are the important factors affecting the activity and stability of phage. The phage BUCT700 showed good stability at 4, 20, 30, 40, and 50℃ incubation for up to 24 h and maintain high titers in the first 6 h under high-temperature conditions (60 and 70℃). In the pH stability experiments, phage BUCT700 had a broad range of pH tolerance (pH 4 to 12) under prolonged incubation conditions and kept high activity at pH 2 for 6 h. The results of temperature and pH tests provide strong support for phage BUCT700 as clinical antimicrobial drugs. Long-term storage at the low temperature is also an important indicator of a drug. The viability of BUCT700 was not affected by long-term storage of up to 5 months at 4℃, which indicated that BUCT700 could be stored as drugs for long periods in hospitals and is equipped for long-distance transport and mass production.

Phage BUCT700 has a short adsorption time with high adsorption and a short multiplication time in the host cells with a large number of phages released. The results of antibacterial activity assessment *in vitro* showed that phage BUCT700 can rapidly kill host bacteria at low MOI and has a good bactericidal ability. Furthermore, phage BUCT700 showed promising efficacy in prophylactic and therapeutic *G. mellonella* larvae models. Single phage dose showed significant efficacy in prophylactic and treatment, respectively. And phage BUCT700 was not toxic for *G. mellonella* larvae. The above results show that phage BUCT700 has

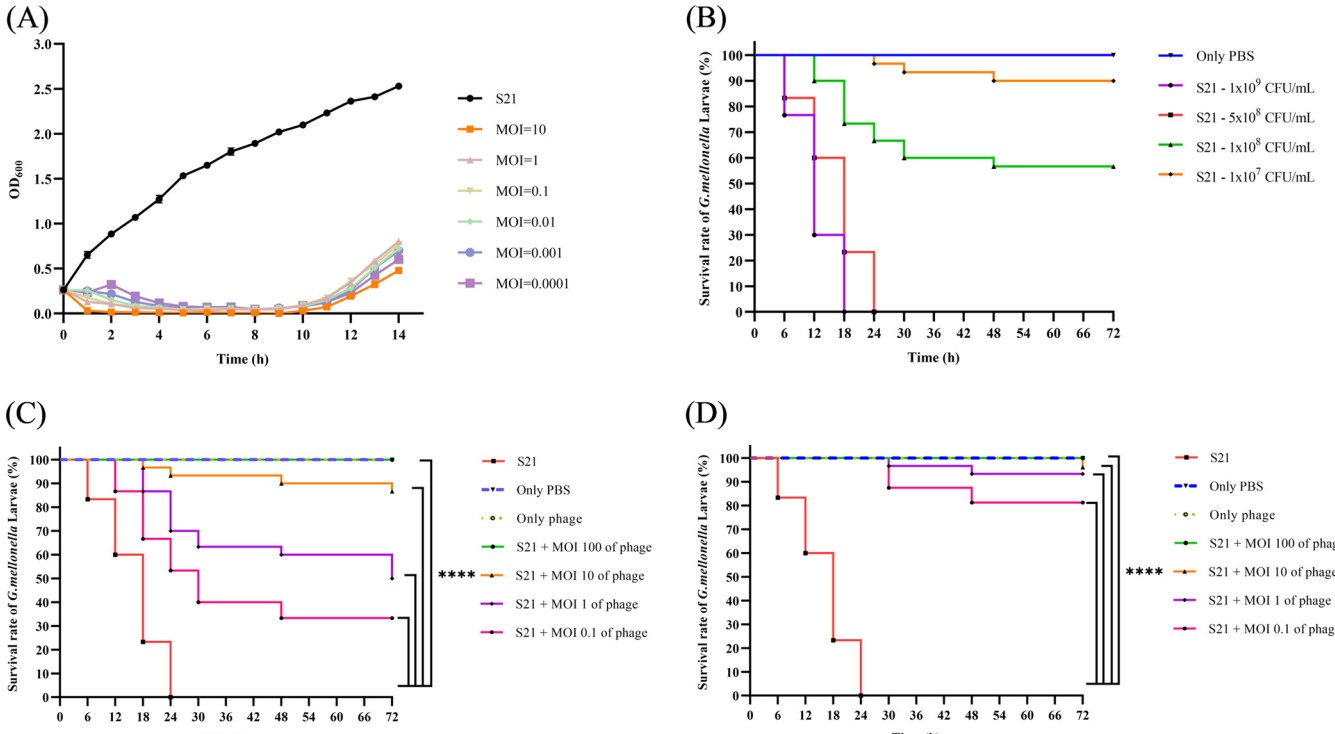

**FIG 8** Therapeutic efficacy in *vitro* and *vivo*. (A) Lytic activity of phage BUCT700 against *S. maltophili* S21 at different MOIs; (B) Survival curve of *G. mellonella* after injecting with different concentrations of *S. maltophili* S21; (C) Survival rate on therapeutic treatment; (D) Survival rate on prophylactic treatment. The log rank (Mantel-Cox) test and the Gehan-Breslow-Wilcoxon test were used on survival curves of *G. mellonella* analysis (C and D) and a *P*-value of <0.05 was considered statistically significant. The symbols are defined as follows: ns: no significant; *, $P < 0.05$; **, $P < 0.01$; ***, $P < 0.001$; and ****, $P < 0.0001$. The results showed that the negative and experimental groups had a significant difference (the log rank [Mantel-Cox] test: $P < 0.0001$; the Gehan-Breslow-Wilcoxon test: $P < 0.0001$), compared with the positive-control group *S. maltophili* S21.

strong potential for clinical application either as a prophylactic or therapeutic agent. Especially as prophylaxis, phage BUCT700 could be considered a prophylactic agent to be added to the treatment regimen. In addition, phage BUCT700 could be used as an instrument surface biocide to reduce the risk of *S. maltophilia* infection caused by instrument retention.

In conclusion, a novel phage BUCT700 with activity against a wide range of ST types of *S. maltophilia* was isolated and characterized. The phage exhibited good tolerance to a broad range of temperature and pH and was stable under long-term storage conditions. High efficacy in prophylactic and therapeutic *G. mellonella* larvae models demonstrates the potential for clinical application of phage BUCT700. Phage BUCT700 is promising to reduce the risk of complications caused by clinical *S. maltophilia* infection and to reduce the difficulty and period of treatment.

## MATERIALS AND METHODS

**Bacterial strain and cultures conditions.** *S. maltophilia* clinical isolate S21 (GenBank accession number: SRX18445116, antibiogram was shown in Table 3) was used as the host of bacteriophage BUCT700. Forty-three clinical *S. maltophilia* strains from different hospitals of china were used for the host spectrum assay of BUCT700. The strains used in this study were cultured in Luria-Bertani (LB) liquid medium or LB agar (1.5% W/V) plates at 37℃ and stored at −80℃ in 50% glycerol (vol/vol) (37).

**Isolation and purification of BUCT700.** Bacteriophage BUCT700 was isolated from untreated sewage samples of the Aviation General Hospital sewer system. *S. maltophilia* S21 was used as the host strain to enrich phages directly from sewage, and phages were isolated by the double-layer plate method (38). In brief, untreated sewage samples were centrifuged and filtered using 0.22-μm filter (Millipore, United States) to remove bacteria and other particles. 500 μL of the filtrate was cocultured with 500 μL of *S. maltophilia* S21 (OD$_{600}$ = 0.6) in 5 mL LB medium, followed by incubation overnight at 37℃ with shaking. The mixed culture was centrifuged at 12,000 × g for 3 min and then the supernatant was filtered with a filter (0.22 μm) to remove residual bacterial cells and debris. The filtrate (2 μL) was spotted on LB medium plate which contained *S. maltophilia* S21. Subsequently, the plate was incubated overnight at 37℃ for acquiring phage plaques.

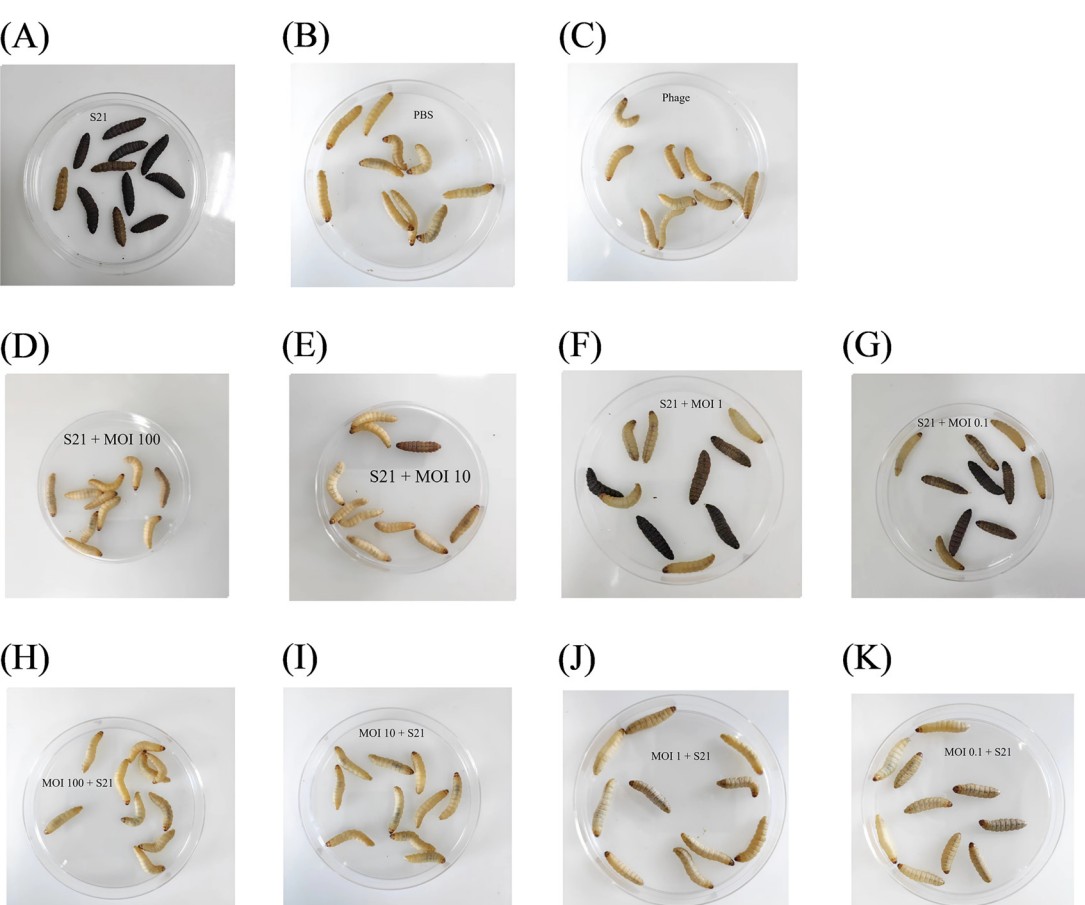

**FIG 9** Morphology of *G. mellonella* larvae. Surviving *G. mellonella* larvae were cream colored without spots, while dead *G. mellonella* larvae were black. (A) Only S21 without phage BUCT700; (B) Only PBS; (C) Only phage; (D-G) injected with S21 followed by phage BUCT700; (H-K) injected with phage BUCT700 followed by S21.

Phages were purified by five successive single-plaque isolation and then further purified by cesium chloride (CsCl) density gradient centrifugation (38, 39).

**Electron microscopy.** To visualize phages, the phage particles were negatively stained with 2% (W/V) phosphotungstic acid (pH = 7.0) for 2 min and examined using a JEM-1200EX transmission electron microscope at an acceleration voltage of 80 kV (40).

**Host range analysis of BUCT700 and multilocus sequence typing (MLST) of *S. maltophilia* strains.** Forty-three clinical *S. maltophilia* strains, which were isolated from the hospital environment, were provided by several hospitals. The phage lysate ($1 \times 10^9$ PFU/mL) was diluted to different titers ($10^2$ to $10^8$ PFU/mL). Then, the host range of phage BUCT700 was tested by spotting 10 $\mu$L of different titers onto the plate of different strains and incubated at 37°C for 8 h. Efficiency of plating (EOP) of tested strains that were positive for the spot test was determined against *S. maltophilia* S21 to identify the ability of phage BUCT700 to reproduce with the help of bacteria (41). Briefly, 100 $\mu$L tested strain was mixed with 100 $\mu$L BUCT700 at an MOI of 1 and 5 mL of LB liquid medium, and then coculture at 37°C for 12 h. After centrifugation (12,000 $\times$ *g*, 5 min) and filtration (0.22 $\mu$m), phage titers were counted by double-layer assay. The EOP of phage BUCT700 was calculated by dividing the phage titers on the tested strain by the phage titers on *S. maltophilia* S21 (42).

Seven pair primers targeting the conserved regions of seven housekeeping genes of *S. maltophilia* were

**TABLE 3** Antibiogram of *Stenotrophomonas maltophilia* S21[a]

| Antibiotic | K-B (mm) | Sensitivity |
|---|---|---|
| Chloramphenicol | | R |
| Ampicillin | | R |
| Polymyxin | | R |
| Kanamycin | | S |
| Levofloxacin | 18 | S |
| Minocycline | 19 | S |
| Sulfamethoxazole | 18 | S |

[a]R, resistant; S, sensitivity.

**TABLE 4** Primers information

| Primer name | Sequence (5′ -3′) |
| --- | --- |
| atpD | F-ATGAGTCAGGGCAAGATCGTTC |
| | R-TCCTGCAGGACGCCCATTTC |
| gapA | F-TGGCAATCAAGGTTGGTATCAAC |
| | R-TTCGCTCTGTGCCTTCACTTC |
| guaA | F-AACGAAGAAAAGCGCTGGTA |
| | R-ACGGATGGCGGTAGACCAT |
| mutM | F-AACTGCCCGAAGTCGAAAC |
| | R-GAGGATCTCCTTCACCGCATC |
| nuoD | F-TTCGCAACTACACCATGAAC |
| | R-CAGCGCGACTCCTTGTACTT |
| ppsA | F-CAAGGCGATCCGCATGGTGTATTC |
| | R-CCTTCGTAGATGAAGCCGGTGTC |
| recA | F-ATGGACGAGAACAAGAAGCGC |
| | R-CCTGCAGGCCCATCGCC |
| PilX | F-cgctctagaactagtggatccATGAAGCCCTCCGTCTCCC |
| | R-gtcgacggtatcgataagcttTCAAAGCTTGAAAACAGTGTCGA |

selected from MLST website (https://pubmlst.org/organisms/stenotrophomonas-maltophilia/primers) (42). The genes included *atpD*, *gapA*, *guaA*, *mutM*, *nuoD*, *ppsA*, and *recA* were subjected to PCR amplification. The amplified products were sent to Beijing Ruibo Xingke Biotechnology Co., Ltd., for bidirectional sequencing. The obtained sequence was submitted to PubMLST database (https://pubmlst.org/) to determine the sequence typing (ST) (43). Primer sequences were shown in Table 4.

**Optimal multiplicity of infection determination.** To determine the optimal MOI of phage BUCT700, the mixture, including BUCT700 and *S. maltophilia* S21 with different MOIs (100, 10, 1, 0.1, 0.01, 0.001, 0.0001, and 0.00001) was added into 10 mL LB liquid medium for overnight culture at 37℃ with 200 rpm shaking. Then the titer of phage was determined by the double-layer plate method and the MOI with the highest phage titer was the optimal MOI (44). The experiments were performed in triplicate.

**Adsorption curve.** *S.maltophilia* S21, S21R and S21R + PilX were mixed with phages at an MOI of 1 and incubated at 37℃ with shaking, respectively. 100 $\mu$L samples were taken, respectively, at 0, 2, 4, 6, 8, 10, 12, 14, 16, and 18 min and suspended in 900 $\mu$L LB liquid medium, followed by centrifugation at 12,000 × $g$ for 5 min. Then, 100 $\mu$L supernatant was taken to determine the titer of free phages in the culture by the soft agar overlay method. The phage adsorption efficiency was calculated with the following equation: (initial phage titer-unabsorbed phage titer in the supernatant)/initial phage titer multiplied by 100% (45).

**One-step growth curve.** *S.maltophilia* isolate S21 cells were infected with phage BUCT700 at an MOI of 1 and allowed to adsorb for 8 min at room temperature. The supernatant was discarded after centrifugation at 4℃ and 12,000 × $g$ for 5 min. The precipitate was resuspended with LB liquid medium and then centrifuged at 4℃ and 12,000 × $g$ for 5 min, with the above steps repeated. The resuspension mixtures were added to 20 mL of LB liquid medium, followed by incubation at 37℃. Samples were taken at 5-min intervals for the first 30 min, and then at 10-minute intervals up to 160 min. The phage titer was determined via the soft agar overlay method. The latent period and burst period were obtained directly from the one-step growth curve.

**The stability determination.** The method of phage BUCT700 stability determination is similar as described by Kusradze et al. but with slight changes (46). Briefly, the phage BUCT700 suspension was incubated at 4, 20, 30, 40, 50, 60, 70℃, and samples were taken at 1, 3, 6, 12, 24 h, respectively. Similarly, for determination of pH stability, the phage solution was incubated at pH 2, 4, 6, 8, 10, 12 at 37℃ and sampling time points as same as the temperature stability experiment. For examining the preservation ability of phage BUCT700 as a formulation, the purified phages were stored at 4℃ for 5 months and the stability of BUCT700 was measured monthly.

In the above-described experiments, the titer of phage BUCT700 was detected by the double-layer plate method and performed three parallel experiments.

**Selecting spontaneous phage-resistant mutants.** To select spontaneous phage-resistant mutant S21R, a modified method was used (47). *S. maltophilia* S21 was mixed with phage BUCT700 at an MOI of 100, giving enough survival pressure to bacteria S21. Following overnight incubation at 37℃, the bacteria-phage mixture was centrifuged at 4℃ and 12,000 × $g$ for 5 min. The precipitate was resuspended with 1 mL of LB liquid medium. 100 $\mu$L resuspension solution was streaked on LB agar plate with high concentrations of phage BUCT700 (1 × 10$^9$ PFU/mL) and grown overnight at 37℃. Then we picked a single colony into 5 mL of LB liquid medium to enrich the bacteria. To verify that the colony is resistant to phage BUCT700, the phage lysate (1 × 10$^9$ PFU/mL) was diluted to different titers (10 to 10$^8$ PFU/mL). Then, the colony was tested by spotting 2 $\mu$L of different titers onto the plate of the colony and incubated at 37℃ for 8 h.

To investigate the receptor binding site for phage BUCT700, we examined the above obtained phage-resistant mutants for impaired adsorption. Phage-resistant bacteria was infected with phage BUCT700 at an MOI of 1 and allowed to adsorb for 8 min at room temperature. Then the mixture was centrifuged at 4℃ and 12,000 × $g$ for 5 min. The supernatant was serially diluted by 1 in 10 into individual wells of 96-well microplates containing 200 $\mu$L of LB medium. Two $\mu$L of the serial dilutions were spotted onto the plate of phage-resistant bacteria. The plate was inverted, incubated at 37℃ overnight, and then examined and photographed. The PBS and *S. maltophilia* S21 were used as control.

**DNA sequencing and mutation analysis of phage-resistant mutant S21R.** The genomic DNA of *S. maltophilia* S21 and phage-resistant mutant S21R was extracted by Bacterial Genomic DNA Extraction kit (Beijing Solarbio Science & Technology Co., Ltd.) and sent to Novogene Bioinformatics Technology Co., Ltd., for the next generation sequencing. The raw sequencing reads of *S. maltophilia* S21 were assembled using SPAdes v3.13.0 (48). And the contigs of *S. maltophilia* S21 were used as the model for mutation analysis. Mutations were identified from next-generation sequencing data of phage-resistant mutant S21R by using *breseq* (https://barricklab.org/twiki/bin/view/Lab/ToolsBacterialGenomeResequencing) (49).

**Molecular genetic methods for identifying adsorbed receptor.** Based on the mutation analysis results, we performed validation of adsorbed receptor complementation on mutant S21R (GenBank accession number: SRX18445116). Complementation of mutant S21R was performed as previously described with some modifications (50). Briefly, the PilX gene was amplified from *S. maltophilia* S21 by colony PCR using primer pairs PilX-F and PilX-R (Table 4). The products were inserted into the pBBR1MCS-2 using conventional cloning techniques, with restriction enzymes BamHI and HindIII (51). The resulting PilX-expressing plasmid was designated pBBR1MCS-2-PilX.

The electroporation of mutant S21R competent cells were prepared as described previously (51). Then 50 $\mu$L of electrocompetent cell was mixed with 3 $\mu$L pBBR1MCS-2-PilX and electroporated using a Bio-Rad GenePulser Xcell. (United States) with pulse controller set at 1.8 KV, 25 $\mu$F, 200 $\Omega$ and 0.1 cm cuvette width. Immediately followed by the cell revitalization, the sample was suspended in 1 mL of LB medium and incubated at 37°C for 2 h and centrifuged at 12,000 $\times$ *g* for 5 min, and discard 900 $\mu$L supernatant and resuspend the rest. The resuspension solution was coated on LB agar plate with kanamycin for incubating at 37°C. Then we picked a single colony into 5 mL of LB liquid medium with kanamycin to enrich the bacteria S21R + PilX. The lysis activity of phage and impaired adsorption were examined using the same methods in the section on Selecting spontaneous phage-resistant mutants.

**DNA sequencing and genome analysis of phage BUCT700.** Phage genomic DNA was extracted by the classical K/SDS method (52, 53). Extracted DNA was quantified using a QubitTM fluorometer, and 100 ng of DNA was used in NEBNext Ultra II FS DNA Library Prep kit (NEB) as manufacturer's instructions. Briefly, the genomic DNA was fragmented, indexed by PCR and purified using AMPure XP beads. Agilent 2100 Bioanalyzer system was used to measure the size distribution of the constructed library fragments, and the library was quantified using the KAPA Library Quantification kits. The pooled DNA library was sequenced on the Illumina NovaSeq and paired-end 2 $\times$ 150 bp reads generated (54).

The raw sequencing data quality was analyzed using the quality control software FastQC v0.11.5 and filtered for low quality reads and adapter regions using Trimmomatic 0.36 with default parameters (55). The generated high-quality reads were assembled using SPAdes v3.13.0 (48). Nucleotide sequence similarities were identified using BLASTn (https://blast.ncbi.nlm.nih.gov/Blast.cgi?PROGRAM=blastn&PAGE_TYPE=BlastSearch&BLAST_SPEC=&LINK_LOC=blasttab&LAST_PAGE=blastp). Potential open reading frames (ORFs) of the phage genome were predicted by using RAST website (https://rast.nmpdr.org/rast.cgi) (56). Then, all predicted ORFs were checked against National Center for Biotechnology Information (NCBI) nonredundant protein database (nr) with an E-value cutoff of $1 \times 10^{-5}$ and interpro (https://www.ebi.ac.uk/interpro/). And molecular mass of proteins encoded by ORFs was determined by Expasy ProtParam online website (https://web.expasy.org/protparam/) (57). The presence of tRNAs in the genome was examined by tRNAscan-SE v.2.0 (http://lowelab.ucsc.edu/cgi-bin/tRNAscan-SE2.cgi) (58). The virulence determinants and the genes involved in antibiotic resistance were determined using the VirulenceFinder (https://cge.food.dtu.dk/services/VirulenceFinder/) and ResFinder (https://cge.food.dtu.dk/services/ResFinder/), respectively (59). Circular genome mapping was created using an in-house python script and retouched by the software Inkscape 0.92.3.0. A comparative analysis of the phage genome with its closest relatives was conducted using Easyfigv2.2.3 at the DNA level. The phylogenetic trees were constructed based on DNA polymerase (ORF22), the terminase small subunit (ORF51) and the terminase large subunit (ORF52). The Neighbor-Joining (NJ) method in Molecular Evolutionary Genetic Analysis (MEGA) v7.0 was used to generate the trees with 1,000 bootstraps.

**Bacterial cell killing assay *in vitro*.** 1 mL of *S. maltophilia* S21 at the early exponential growth phase was inoculated into 10 mL LB medium. When the $OD_{600}$ of the cultures about 0.2, phage BUCT700 was infected with MOI of 10, 1, 0.1, 0.01, 0.001, and 0.0001. The uninfected culture was used as the positive control. 200 $\mu$L samples were collected hourly and measured $OD_{600}$ with nanodrop (Thermo scientific).

**Assessment of therapeutic efficacy in *Galleria mellonella* larvae.** To provide a reference for clinical use, *G. mellonella* larvae (Huiyude Biotech Company, Tianjin, China) were used as a model for assessing the efficacy of phages against infection. Active *G. mellonella* larvae with approximately 300 mg weight were selected. In order to select the appropriate infectious dose, 5 $\mu$L different concentrations of S21 cells were injected into the last left pro-leg using a microsyringe (Gaoge, Shanghai, China) (60).

To determine the therapeutic efficacy of phage, freshly cultured *S. maltophilia* S21 solution was washed with phosphate-buffered saline (PBS) and then diluted to the $5 \times 10^8$ CFU/mL. 5 $\mu$L *S. maltophilia* S21 cells ($5 \times 10^8$ CFU/mL) were injected into the last left pro-leg. After 1 h of infection, 5 $\mu$L phage suspension at MOI of 100, 10, 1, 0.1 were, respectively, injected on the opposite side to the bacterial injection site, followed by incubation at 37°C in darkness. For the prophylactic treatment model of phage, *G. mellonella* larvae were injected with phage followed by bacteria infection. Briefly, 5 $\mu$L phage suspension at MOI of 100, 10, 1, 0.1 were, respectively, injected into the last left pro-leg. After 1 h of phage treatment, 5 $\mu$L *S. maltophilia* S21 cells ($5 \times 10^8$ CFU/mL) were injected on the opposite side of the larvae to the phage inoculation, followed by incubation at 37°C in darkness.

The positive-control group of *G. mellonella* larvae was infected with *S. maltophilia* S21 and treated with PBS. The *G. mellonella* larvae were, respectively, injected with PBS and phage with an MOI of 100 as the negative controls. A group of 10 *G. mellonella* larvae was used and each group was used to perform three parallel

experiments. The survival rate of *G. mellonella* larvae was recorded at 6-h intervals up to 72 h. The dark larvae that did not respond to physical contact were marked deceased.

**Statistic statements.** All data were analyzed using GraphPad Prism 8.0.1 and expressed as means and standard deviation values. One-way analysis of variance was used in Fig. 2A. And survival curves of *G. mellonella* (Fig. 8C and 8D) were analyzed by the log rank (Mantel-Cox) test or the Gehan-Breslow-Wilcoxon test. A *P*-value g of $< 0.05$ was considered statistically significant. The symbols are defined as follows: ns, no significant; *, $P < 0.05$; **, $P < 0.01$; ***, $P < 0.001$; and ****, $P < 0.0001$.

**Data availability.** The complete genome sequence of phage BUCT700 with annotations was submitted to the GenBank database under the accession number OM735686. The original sequencing data of *S. maltophilia* S21 and S21R were submitted to the NCBI databases under the accession number SRX18445116 and SRX18445117, respectively.

All dates generated for this study are included in the article.

## ACKNOWLEDGMENTS

Yahao Li: resources, data curation, writing-original draft, investigation. Mingfang Pu, Pengjun Han: data curation, investigation. Mengzhe Li, Xiaoping An, Lihua Song: investigation, validation. Huahao Fan, Zeliang Chen: supervision, writing-review & editing. Yigang Tong: conceptualization, supervision.

This research was funded by National Key Research and Development Program of China (No. 2018YFA0903000), Innovation & Transfer Fund of Peking University Third Hospital (No. BYSYZHKC2022114), Funds for First-class Discipline Construction (No. XK1805, No. XK1803-06), Inner Mongolia Key Research and Development Program (No. 2019ZD006), NSFC-MFST project (China-Mongolia) (No. 31961143024), Fundamental Research Funds for Central Universities (No. BUCTRC201917, BUCTZY2022).

We have no conflicts of interest to declare.

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
