## [Reviewer comments · Microbiology Spectrum]

Microbiology Spectrum

Efficacy in *Galleria mellonella* larvae and application potential assessment of a new bacteriophage BUCT700 extensively lyse *Stenotrophomonas maltophilia*

Yahao Li, Mingfang Pu, Pengjun Han, Mengzhe Li, Xiaoping An, Lihua Song, Huahao Fan, Zeliang Chen, and Yi-Gang Tong

Corresponding Author(s): Huahao Fan, Beijing University of Chemical Technology

Review Timeline:

Submission Date:	October 4, 2022
Editorial Decision:	November 2, 2022
Revision Received:	December 21, 2022
Accepted:	January 8, 2023

Editor: Xiaoyu Tang

Reviewer(s): The reviewers have opted to remain anonymous.

Transaction Report:

DOI: <https://doi.org/10.1128/spectrum.04030-22>

November 2, 2022

Dr. Huahao Fan
Beijing University of Chemical Technology
No.98 Zi-Zhu-Yuan-Lu
Haidian District
Beijing 100029
China

Re: Spectrum04030-22 (Clinical application potential and efficacy in *Galleria mellonella* larvae assessment of a new bacteriophage BUCT700 extensively lyse *Stenotrophomonas maltophilia*)

Dear Dr. Huahao Fan:

Link Not Available

Sincerely,

Xiaoyu Tang

Journals Department
Reviewer comments:

Reviewer #1 (Comments for the Author):

The authors provide a bacteriophage named BUCT700 which can lyse multiple clinical ST species of *Stenotrophomonas maltophilia*, which is a clinical challenge. The safety and therapy potential of phage BUCT700 for clinical applications were evaluated by genomic level analysis and stability experiments. The authors used a *Galleria mellonella* model to examine the efficacy and safety of treatment with phage BUCT700 in vivo. Interestingly, phage BUCT700 was able to provide effective protection to *Galleria mellonella*. This is a rigorous and interesting work. The authors evaluated the potential for clinical application of phage BUCT700 and proposed various ways for clinical use of phage BUCT700 in the discussion section, which

provided new ideas for clinical application of phage therapy.

Title: "Clinical application potential" is over-interpreted. All the experiments are basic research but not pre-clinical studies.

The manuscript needs English editing. There have some grammatical errors in this manuscript that should be corrected carefully and thoroughly.

1) Italicized words

Line 27, correct it as "in vivo".

Line 78, correct it as "S.maltophilia". Note that Latin words should usually be printed in italics. Authors should check and correct all Latin words in the article.

Line 215-216, correct it as "in vitro and in vivo".

2) Line 28, please replace "S.maltophilia infected" with "S.maltophilia-infected".

3) Line 32, please remove "the".

4) Line 66, please replace "for" with "of".

5) Line 174, please replace "DNA binding" with "DNA-binding".

6) Line 233, please replace "have" with "has".

7) The article "an" should be used in front of MOI instead of "a".

1. Line 29: I strongly suggest removing "G.mellonella larvae".

2. Line 79-81: The meaning of "phages" is not clear. I suggest replacing "phages" with " S.maltophilia phages".

3. Line 162: please remove "time".

4. Line 224-225: I suggest replacing "phage" with "phage BUCT700".

How to determine death of *Galleria mellonella*? How was the dose chosen? Provide more details and the references. If you have done different gradient dose examinations, please explain and provide the corresponding survival curve.

The the accession number should be listed in the "Data availability"

Statistic statements are missing in the METHODS section.

The figure legends are not very informative.

Reviewer #2 (Comments for the Author):

Infections caused by multidrug-resistant *S. maltophilia* have emerged as a serious challenge in clinical practice, Li et al. isolated a phage BUCT700 that can be effective for most clinical *S. maltophilia* isolates. The author well characterized this phage including one-step growth curve, replication kinetics, stability, host range, genome analysis and in vivo efficacy in a *Galleria mellonella* (*G.mellonella*) larvae model. This work demonstrated the potential of phage BUCT700 for future clinical utilization. However, the major concerns for this reviewer are the novelty and significance of this study to the readers of this journal. The data are not well presented in this manuscript. Figures are redundant and can be merged. The manuscript needs further improvement with the following suggestions:

1. Since the host bacteria S21 is not a standard strain but a clinical strain, the author should provide the characterization of this strain including the genome information, antibiogram and etc.

2. In the section of results "Genomic analysis and annotation of phage BUCT700", although authors try to describe the function of each annotated genes, this seems wordy, since these genes are general founded in other phages as well, the authors should focus on the specificity of the phage BUCT700.

3. Line 178, the author mixed the definition of endolysin and lysozyme, indeed the endolysin including the lysozyme. The author should check the annotation of the whole phage genome by interpro, HMMER or other tools for more accurate results.

4. Line 188, this sentence is not well written, change to "The lytic activity of phage BUCT700 with *S. maltophilia* S21"

5. The in vivo experimental result needs statistic analysis.

6. Figure 7 B and C is not well drawn, control group and phage only group is not separated in the figure. Please make a clearer marker.

7. Since the author found the phage BUCT700 is a wide host-range phage, it's a pity that the author did not investigate the receptor binding site for this phage, I suggest the author studying the receptor binding site for this phage.

8. All the sequences should be deposited in a public database with accession no.

Staff Comments:

Preparing Revision Guidelines

To submit your modified manuscript, log onto the eJP submission site at <https://spectrum.msubmit.net/cgi-bin/main.plex>. Go to

Author Tasks and click the appropriate manuscript title to begin the revision process. The information that you entered when you first submitted the paper will be displayed. Please update the information as necessary. Here are a few examples of required updates that authors must address:

Please return the manuscript within 60 days; if you cannot complete the modification within this time period, please contact me. If you do not wish to modify the manuscript and prefer to submit it to another journal, please notify me of your decision immediately so that the manuscript may be formally withdrawn from consideration by Microbiology Spectrum.

北京化工大学

Beijing Advanced Innovation Center for
Soft Matter Science and Engineering
Beijing University of Chemical Technology
Beijing, 100029, China

Dr. Huahao Fan

Email: fanhuahao@mail.buct.edu.cn

December 18, 2022

Editor

Dr. Xiaoyu Tang

Microbiology Spectrum

RE: Decision on Spectrum04030-22

**Clinical application potential and efficacy in *Galleria mellonella* larvae
assessment of a new bacteriophage BUCT700 extensively lyse *Stenotrophomonas
maltophilia***

Dear Dr. Xiaoyu Tang,

Thank you for your email of November 2, 2022 and for your as well as the reviewers' insightful comments concerning our manuscript. We are very grateful for sending us the revision package on our manuscript Spectrum04030-22. According to your suggestions and the reviewers' comments, we have revised the manuscript.

This manuscript has been carefully revised to address the comments of referees and in the editorial checklist. Following is a detailed explanation of how we address the issues raised by reviewers. Looking forward to hearing from you soon.

Sincerely yours,

Huahao Fan, Ph.D, Associate Professor

College of Life Science and Technology,
Beijing University of Chemical Technology,
Beijing 100089, China.

Email: fanhuahao@mail.buct.edu.cn

Incl.: Point-by-Point Responses to the Reviewer's Comments

Reviewer #1 (Comments for the Author):

The authors provide a bacteriophage named BUCT700 which can lyse multiple clinical ST species of *Stenotrophomonas maltophilia*, which is a clinical challenge. The safety and therapy potential of phage BUCT700 for clinical applications were evaluated by genomic level analysis and stability experiments. The authors used a *Galleria mellonella* model to examine the efficacy and safety of treatment with phage BUCT700 *in vivo*. Interestingly, phage BUCT700 was able to provide effective protection to *Galleria mellonella*. This is a rigorous and interesting work. The authors evaluated the potential for clinical application of phage BUCT700 and proposed various ways for clinical use of phage BUCT700 in the discussion section, which provided new ideas for clinical application of phage therapy.

Response: Thanks for your approbation of this manuscript, and we appreciate your constructive suggestions. We have addressed the issues you pointed out to improve the quality of our manuscript.

Title: "Clinical application potential" is over-interpreted. All the experiments are basic research but not pre-clinical studies.

Response: Thank you for your kind reminder. We have removed "Clinical" from the title and changed the title to "Efficacy in *Galleria mellonella* larvae and application potential assessment of a new bacteriophage BUCT700 extensively lyse *Stenotrophomonas maltophilia*".

The manuscript needs English editing. There have some grammatical errors in this manuscript that should be corrected carefully and thoroughly.

Response: Thank you for your constructive suggestions regarding our manuscript, we carefully checked the whole manuscript for any typos and polished all sections in current manuscript.

1) Italicized words

Line 27, correct it as "in vivo".

Response: Thank you for your suggestion, we have changed the sentence to “*In vivo*, BUCT700 significantly increased the survival rate of *S.maltophilia*-infected *Galleria mellonella* (*G.mellonella*) larvae from 0% to 100% within 72 h, especially in the prophylactic model.”. (Line 30-32)

Line 78, correct it as "S.maltophilia". Note that Latin words should usually be printed in italics. Authors should check and correct all Latin words in the article.

Response: Thank you for your suggestion, we carefully checked the whole manuscript and correct all Latin words.

“However, most of the existing studies about *S.maltophilia* phages rested on genomics and their physiological characteristics.” (Line 81-82)

Line 215-216, correct it as "in vitro and in vivo".

Response: Thank you for your suggestion, we have changed the sentence to “This study provides a novel phage BUCT700 and analyzes its clinical application potential *in vitro* and *in vivo*.”. (Line 216-217)

2) Line 28, please replace "S.maltophilia infected" with "S.maltophilia-infected".

Response: Thank you for your suggestion, we have changed the sentence to “*In vivo*, BUCT700 significantly increased the survival rate of *S.maltophilia*-infected *Galleria mellonella* (*G.mellonella*) larvae from 0% to 100% within 72 h in the therapeutic model.”. (Line 30-32)

3) Line 32, please remove "the".

Response: Thank you for your suggestion, we have changed the sentence to “*In vivo*, BUCT700 significantly increased the survival rate of *S.maltophilia*-infected *Galleria mellonella* (*G.mellonella*) larvae from 0% to 100% within 72 h in the therapeutic model.”. (Line 30-32)

4)Line 66, please replace "for" with "of".

Response: Thank you for your suggestion, we have changed the sentence to “Currently, only a few antibiotics like Sulfamethoxazole-Trimethoprim (SXT) and Levofloxacin can be used for clinical treatment of *S.maltophilia* infection.”. (Line 68-69)

5)Line 174, please replace "DNA binding" with "DNA-binding".

Response: Thank you for your suggestion. Based on the comments of reviewer 2, we focused on the specificity of the phage BUCT700 and deleted this sentence.

6)Line 233, please replace "have" with "has".

Response: Thank you for your suggestion, we have changed the sentence to “The phage BUCT700 has a high nucleotide sequence similarity to *S.maltophilia* phage BUCT598 and *S.maltophilia* phage P15, indicating BUCT700 may has similar life cycles with them.”. (Line 236-238)

7)The article "an" should be used in front of MOI instead of "a".

Response: Thank you for your suggestion, we carefully checked the whole manuscript and correct the article before MOI.

“The group of only injected PBS and only injected phage at an MOI of 100 survived for 72 h, indicating that either PBS or phage did not cause any death of *G.mellonella* larvae.” (Line 197-199)

“Briefly, 100 μ L tested strain was mixed with 100 μ L BUCT700 at an MOI of 1 and 5 mL of LB liquid medium, and then co-culture at 37 °C for 12 h.” (Line 306-307)

“*S.maltophilia* S21, S21R and S21R + PilX were mixed with phages at an MOI of 1 and incubated at 37 °C with shaking, respectively.” (Line 325-326)

“*S.maltophilia* isolate S21 cells were infected with phage BUCT700 at an MOI of 1 and allowed to adsorb for 8 min at room temperature.” (Line 333-334)

“The *G.mellonella* larvae were respectively injected with PBS and phage with an MOI of 100 as the negative controls.” (Line 446-447)

1.Line 29: I strongly suggest removing "G.mellonella larvae".

Response: Thank you for your suggestion, we have changed the sentence to “*In vivo*, BUCT700 significantly increased the survival rate of *S.maltophilia*-infected *Galleria mellonella* (*G.mellonella*) larvae from 0% to 100% within 72 h in the therapeutic model.”. (Line 30-32)

2.Line 79-81: The meaning of "phages" is not clear. I suggest replacing "phages" with " *S.maltophilia* phages".

Response: Thank you for your suggestion, we have changed the sentence to “Current studies are deficient in systematic assessment of the clinical therapeutic potential of *S.maltophilia* phages and lack data on treatment effects *in vivo* through animal models of infection.”. (Line 82-84)

3.Line 162: please remove "time".

Response: Thank you for your suggestion. Based on the comments of reviewer 2, we focused on the specificity of the phage BUCT700 and deleted this sentence.

4.Line 224-225: I suggest replacing "phage" with "phage BUCT700".

Response: Thank you for your suggestion, we have changed the sentence to “The results of temperature and pH tests provide strong support for phage BUCT700 as clinical antimicrobial drugs.”. (Line 253-254)

How to determine death of *Galleria mellonella*? How was the dose chosen? Provide more details and the references. If you have done different gradient dose examinations, please explain and provide the corresponding survival curve.

Response: Thank you for your question, we have added more details and references to the section on Assessment of therapeutic efficacy in *Galleria mellonella* larvae.

We refer to the review of Catherine Jia-Yun Tsai et al. to determine the survival of *Galleria mellonella* larvae (DOI: [10.1080/21505594.2015.1135289](https://doi.org/10.1080/21505594.2015.1135289)). Surviving *G.mellonella* larvae were cream colored without spots, while dead *G.mellonella* larvae were black. Melanization typically starts with distinctive black spots on the cream colored larvae. Complete melanization (black larvae) correlates with death of the larvae soon after.

We refer to the research of José Luis Insua et al. to choose the dose of *S.maltophilia* S21 (DOI: [10.1128/IAI.00391-13](https://doi.org/10.1128/IAI.00391-13)). In order to select the appropriate infectious dose, 5 μ L different concentrations of S21 cells were injected into the last left pro-leg using a microsyringe. Then we chose a dose of 5×10^8 CFU/mL which caused significant mortality at 24 h (refer to the high rate of mortality caused by *S.maltophilia* in clinical). The survival curve of *G.mellonella* after injecting with different concentrations of *S.maltophili* S21 was shown in Figure 8B.

(Line 192-196; 432-434; 449-450; 518-520)

Figure 8B. Survival curve of *G.mellonella* after injecting with different concentrations of *S.maltophili* S21.

The accession number should be listed in the "Data availability"

Response: Thank you for your kind reminder, we have listed the accession numbers in the section of Data availability.

“ The complete genome sequence of phage BUCT700 with annotations was submitted to the GenBank database under the accession number OM735686 (<https://www.ncbi.nlm.nih.gov/nuccore/OM735686>). The original sequencing data of *S.maltophilia* S21 and S21R were submitted to the NCBI databases under the accession number SRX18445116 (<https://www.ncbi.nlm.nih.gov/sra/?term=SRX18445116>) and SRX18445117 (<https://www.ncbi.nlm.nih.gov/sra/?term=SRX18445117>), respectively.” (Line 458-463)

Statistic statements are missing in the METHODS section.

Response: Thank you for your kind reminder, we have added the “Statistic statements” to METHODS section.

“Statistic statements

All data were analyzed using GraphPad Prism 8.0.1 and expressed as means and standard deviation values. One-way analysis of variance was used in Figure 2A. And survival curves of *G.mellonella* (Figure 8C and 8D) were analyzed by the log rank (Mantel-Cox) test or the Gehan-Breslow-Wilcoxon test. A P - value of < 0.05 was considered statistically significant. The symbols are defined as follows: ns: no significant; *P < 0.05; **P < 0.01; ***P < 0.001; and ****P < 0.0001.” (Line 451-456)

The figure legends are not very informative.

Response: Thank you for your suggestion, we have supplemented the figure legends to provide a concise and precise explanation of the figures to the readers. (Line 482-485; 488-490; 492-503; 515-516; 518-527)

Reviewer #2 (Comments for the Author):

Infections caused by multidrug-resistant *S. maltophilia* have emerged as a serious challenge in clinical practice, Li et al. isolated a phage BUCT700 that can be effective for most clinical *S. maltophilia* isolates. The author well characterized this phage including one-step growth curve, replication kinetics, stability, host range, genome analysis and in vivo efficacy in a *Galleria mellonella* (*G.mellonella*) larvae model. This work demonstrated the potential of phage BUCT700 for future clinical utilization. However, the major concerns for this reviewer are the novelty and significance of this study to the readers of this journal. The data are not well presented in this manuscript. Figures are redundant and can be merged. The manuscript needs further improvement with the following suggestions:

Response: Thanks for your approbation of this manuscript, and we appreciate your constructive suggestions. We have addressed the issues you pointed out to improve the quality of our manuscript.

The main novelty and significance of this study to the readers of this journal are mainly on the following two aspects:

1. For the first time, we investigated the effect of phages as prophylactic agents in *S.maltophilia*-related infection models and found that phages as prophylactic agents provided better protection against *S.maltophilia* infections compared with the treatment group.

***S.maltophilia* mainly causes infectious complications in the clinical setting and has proven itself a formidable pathogen in the setting of other comorbid conditions (DOI: [10.1097/INF.00000000000003633](https://doi.org/10.1097/INF.00000000000003633); DOI: [10.1093/ofid/ofab644](https://doi.org/10.1093/ofid/ofab644)). Clinically, *S.maltophilia* often causes mixed infections with *Pseudomonas aeruginosa*, *Klebsiella pneumoniae*, and *Acinetobacter baumannii* (DOI: [10.3389/fmed.2021.808391](https://doi.org/10.3389/fmed.2021.808391)). The inability to accurately diagnose the cause of the disease has prevented precise treatment of *S.maltophilia* infections in the first place and exacerbates the disease. And *S.maltophilia* is known to produce**

polymicrobial infections commonly in the respiratory tract in patients with cystic fibrosis as a cocolonizer with *Pseudomonas aeruginosa*, as reported by Brooke JS (DOI: 10.1128/CMR.00019-11). Furthermore, a study from Taiwan reported 14 burns patients with *S.maltophilia* bacteraemia and four deaths in association with polymicrobial sepsis, which also occurs in cancer treatment (DOI: 10.1016/j.burns.2005.08.016; DOI: 10.1097/INF.00000000000003633). The most common manifestation of *S.maltophilia* infections was pneumonia in clinical, which is one of the reasons why it is difficult to be detected and causes deterioration of the disease (DOI: 10.1136/bcr-2021-242670). Patients with severe pneumonia are often indistinguishable from *S.maltophilia* co-infection based on signs, symptoms, physical findings, and radiographic findings (DOI: 10.1016/j.ajic.2021.06.005). There have been 12 cases of co-infection with *S.maltophilia* and COVID-19 reported in the research of Siripen Kanchanasuwan et al. (DOI: 10.3390/jcm11113085). In addition to the aforementioned challenges of *S.maltophilia* that are difficult to diagnose, this emphasizes that *S.maltophilia* is not just a mere colonizer and can impact many other organ systems besides the respiratory system, as reported by Zaryab Umar et al. (DOI: 10.7759/cureus.23541). To sum up, we believe that it is imperative to prevent *S.maltophilia* infection.

Considering its intrinsic resistance to clinically used antibiotics such as penicillins, cephalosporins, carbapenems, aminoglycosides, and macrolides, only a few antibiotics like Sulfamethoxazole-Trimethoprim (SXT) and Levofloxacin can be used for *S.maltophilia* infection. Widespread use of antibiotics as prophylactic agents can lead to the emergence of multiple all-drug-resistant super-bacteria, resulting in serious clinical challenges. Therefore antibiotics are not suitable as prophylactic agents. As bacterial viruses, phages feature exquisite host specificity and do not disrupt microbiota *in vivo* as well as lack major end-organ damage. In addition, phages can also replicate and produce themselves within the host bacteria, which prolongs the effectiveness of phage agents. Using phages as prophylactic agents is a promising practice. However,

there is no report using phages as prophylactic agents against *S.maltophilia* infection. In this study, we investigated for the first time the effect of phages as prophylactic agents in *S.maltophilia*-related infection models and found that phages as prophylactic agents provided better protection against *S.maltophilia* infections compared with the treatment group (Figure 8C and 8D). The results of evaluating phage BUCT700 as prophylactic agents showed that the survival rate of *G.mellonella* larvae was 100% at an MOI of 100, 96.15% at an MOI of 10, 93.33% at an MOI of 1, and 81.25% at an MOI of 0.1, respectively (Figure 8D). In summary, we believe that phage BUCT700 could be considered as a prophylactic agent to be added to the treatment regimen.

Figure 8C. Survival rate on therapeutic treatment.

Figure 8D. Survival rate on prophylactic treatment.

2. For the first time, we found the Type IV fimbrial biogenesis protein PilX is an adsorption receptor of *S.maltophilia* phages, which extends the known phage receptors of *S.maltophilia*.

Among the *S.maltophilia* phages characterized to data, the PilA, PilT, PilE or TonB were used as adsorption receptors, while PilX was not reported (DOI: [10.3389/fmicb.2020.01358](https://doi.org/10.3389/fmicb.2020.01358); DOI: [10.3390/ijms21176338](https://doi.org/10.3390/ijms21176338); DOI: [10.1186/s12864-019-5674-5](https://doi.org/10.1186/s12864-019-5674-5); DOI: [10.3389/fmicb.2022.906961](https://doi.org/10.3389/fmicb.2022.906961); DOI: [10.3390/v10060338](https://doi.org/10.3390/v10060338)). In this study, we found the Type IV fimbrial biogenesis protein PilX is an adsorption receptor of *S.maltophilia* phage BUCT700, which extends the known phage receptors of *S.maltophilia* (Figure 4). PilX highly conserved in bacterial genome was implicated as key promoters of pilus assembly on the cell surface and played a key role in pathogenesis in a variety of bacterial species (DOI: [10.1016/j.vaccine.2011.07.060](https://doi.org/10.1016/j.vaccine.2011.07.060)). Based on the research by Victoria A Marko et al, mutation or loss of PilX had reduced virulence and motility of bacteria (DOI: [10.1371/journal.ppat.1007074](https://doi.org/10.1371/journal.ppat.1007074)). Furthermore, PilX is a pilus-associated protein essential for bacterial aggregation which confers a powerful biofilm-forming ability on *S.maltophilia* (DOI: [10.1111/j.1365-2958.2004.04372.x](https://doi.org/10.1111/j.1365-2958.2004.04372.x)). Bacteriophage BUCT700, which uses PilX as an adsorbed receptor, is able to extensively infest clinical *S.maltophilia* and produces survival pressure causing changes to PilX resulting in reduced virulence of *S.maltophilia*. Therefore, we believe that BUCT700 has good potential as clinical antimicrobial agents.

Figure 4. Adsorption receptor identification of BUCT700. Bacteriophage BUCT700 uses Type IV fimbrial biogenesis protein PiIX as an adsorption receptor to infect *S.maltophilia* S21. (A) Compared to *S.maltophilia* S21, phage-resistant mutant S21R produced significantly impaired adsorption, whereas the complementary strain S21R + PiIX could be re-adsorbed by phage BUCT700. (B) Phage BUCT700 can rapidly adsorb *S.maltophilia* S21, whereas BUCT700 cannot adsorb phage-resistant mutant S21R. When complementation of phage-resistant mutant S21R with the PiIX gene was performed, phage BUCT700 restores adsorption to the complementary strain S21R + PiIX. (C) *S.maltophilia* S21 is susceptible to BUCT700 infection, whereas phage-resistant mutant S21R is resistant to phage infection, but the complementary strain S21R + PiIX restores phage infection to *S.maltophilia* S21 levels. Data are shown as the mean \pm SD.

1. Since the host bacteria S21 is not a standard strain but a clinical strain, the author should provide the characterization of this strain including the genome information, antibiogram and etc.

Response: Thank you for your kind reminder, we have added the genome sequencing data and antibiogram of *S.maltophili* S21. The original sequencing data of *S.maltophilia* S21 was submitted to the GenBank database under the accession number SRX18445116 and the antibiogram of *S.maltophilia* S21 was shown in Table 3.

“The original sequencing data of *S.maltophilia* S21 and S21R were submitted to the GenBank database under the accession number SRX18445116 (<https://www.ncbi.nlm.nih.gov/sra/?term=SRX18445116>) and SRX18445117 (<https://www.ncbi.nlm.nih.gov/sra/?term=SRX18445117>), respectively.” (Line 460-463)

2. In the section of results "Genomic analysis and annotation of phage BUCT700", although authors try to describe the function of each annotated genes, this seems wordy, since these genes are general founded in other phages as well, the authors should focus on the specificity of the phage BUCT700.

Response: Thank you for your suggestion, we have modified this section and focused on the specificity of the phage BUCT700. It is worth noting that BUCT700 has many genes encoded exonucleases (ORF 24, 25 and 26) and tail fiber (ORF 46, 47 and 49), significantly more than other *S.maltophilia* phages. Exonuclease can repair DNA and ensure rapid and correct DNA synthesis indicating that phage BUCT700 has a precise DNA synthesis system to adapt to the needs of rapid replication. These three exonucleases (ORF 24, 25 and 26) of phage BUCT700 make it difficult to carry external virulence and drug resistance genes, improving the safety of its clinical use. Tail fiber was used for recognizing host receptor proteins during the infestation, and more tail fiber proteins may provide a variety the host receptors, which may contribute to its rapid adsorption and broad host spectrum. (Line 162-175; 244-247)

3.Line 178, the author mixed the definition of endolysin and lysozyme, indeed the endolysin including the lysozyme. The author should check the annotation of the whole phage genome by interpro, HMMER or other tools for more accurate results.

Response: Thank you for your kind reminder, we have checked the annotation of the whole phage genome by interpro (<https://www.ebi.ac.uk/interpro/>). We have changed the ORF54 to SAR endolysin (shown in Figure 5).

Figure 5. Schematic diagram of the genome of bacteriophage BUCT700. All predicted ORFs are represented by arrows, where the positive direction is in the outer ring and the negative direction is in the inner ring. Red, lysis; orange, transcription; deeply green, structure; brown, packaging; deeply blue, replication; grey, uncharacterized. Yellow represents greater than the average GC content, and the light green represents the opposite. Purple represents $G-C/G+C > 0$, light blue represents < 0 .

4.Line 188, this sentence is not well written, change to "The lytic activity of phage BUCT700 with *S. maltophilia* S21"

Response: Thank you for your suggestion, we have changed the sentence to “The lytic activity of phage BUCT700 with *S.maltophilia* S21 was determined at different MOIs for 14 h with uninfected *S.maltophilia* S21 as a control.”. (Line 183-184)

5.The in vivo experimental result needs statistic analysis.

Response: Thank you for your suggestion, we have added statistic analysis *in vivo* experimental results. The log rank (Mantel-Cox) test and the Gehan-Breslow-Wilcoxon test were used on survival curves of *G.mellonella* analysis (Figure 8C and 8D) and a P - value of < 0.05 was considered statistically significant. The symbols are defined as follows: ns: no significant; *P < 0.05; **P < 0.01; ***P < 0.001; and ****P < 0.0001. The results showed that the negative and experimental groups had a significant difference (the log rank (Mantel-Cox) test: P < 0.0001; the Gehan-Breslow-Wilcoxon test: P < 0.0001), compared to the positive control group *S.maltophili* S21. (Line 455-458; 522-527)

Figure 8. (C)

(D)

Figure 8. (C) Survival rate on therapeutic treatment; (D) Survival rate on prophylactic treatment. The log rank (Mantel-Cox) test and the Gehan-Breslow-Wilcoxon test were used on survival curves of *G.mellonella* analysis (C and D) and a P - value of < 0.05 was considered statistically significant. The symbols are defined as follows: ns: no significant; *P < 0.05 ; **P < 0.01 ; ***P < 0.001 ; and ****P < 0.0001 . The results showed that the negative and experimental groups had a significant difference (the log rank (Mantel-Cox) test: P < 0.0001 ; the Gehan-Breslow-Wilcoxon test: P < 0.0001), compared to the positive control group *S.maltophili* S21.

6. Figure 7 B and C is not well drawn, control group and phage only group is not separated in the figure. Please make a clearer marker.

Response: Thank you for your suggestion, we have made a clearer marker of negative group and phage only group to separate in Figure 8C and 8D.

Figure 8C. Survival rate on therapeutic treatment.

Figure 8D. Survival rate on prophylactic treatment.

7. Since the author found the phage BUCT700 is a wide host-range phage, it's a pity that the author did not investigate the receptor binding site for this phage, I suggest the author studying the receptor binding site for this phage.

Response: Thank you for your suggestion, we have investigated the receptor binding site for phage BUCT700.

To select spontaneous phage-resistant mutant S21R, the bacterial S21-phage mixture was streaked on LB agar plate with high concentrations of phage BUCT700 (1×10^9 PFU/mL) and grown overnight at 37 °C. Then we used the

phage-resistant mutant S21R which produced significantly impaired adsorption to investigate the receptor binding site for phage BUCT700. Mutations of phage-resistant mutant S21R were identified from next-generation sequencing data by using *breseq* (<https://barricklab.org/twiki/bin/view/Lab/ToolsBacterialGenomeResequencing>).

The results of *breseq* showed that one base mutation in the PilX gene (AGC → AGA) compared to S21, resulted in a mutation from arginine to serine. To verify the results of mutation analysis, we constructed a pBBR1MCS-2-PilX plasmid and electrically transferred it into phage-resistant mutant S21R to obtain the complementary strain S21R + PilX. The results of plaque and adsorption assays on phage-resistant mutant S21R and complementary strain S21R + PilX showed that BUCT700 no longer adsorbed and lysed S21R, whereas the complementary strain S21R + PilX regained the sensitivity to BUCT700, indicating the Type IV fimbrial biogenesis protein PilX is an adsorption receptor of phage BUCT700 (Figure 4). PilX is highly conserved in bacterial genome, was implicated as key promoters of pilus assembly on the cell surface and played a key role in pathogenesis in a variety of bacterial species. Based on the research by Victoria A Marko et al, mutation or loss of PilX had reduced virulence and motility of bacteria (DOI: [10.1371/journal.ppat.1007074](https://doi.org/10.1371/journal.ppat.1007074)). Furthermore, PilX is a pilus-associated protein essential for bacterial aggregation which confers a powerful biofilm-forming ability on *S.maltophilia*. Bacteriophage BUCT700, which uses PilX as an adsorbed receptor, is able to extensively infest clinical *S.maltophilia*, and produces survival pressure causing changes to PilX resulting in reduced virulence of *S.maltophilia*.

The corresponding content of the adsorption receptor identification has been supplemented in the section Abstract, Results, Discussion, Materials and Methods. (Line 25-26; 131-141; 223-232; 327-328; 353-396; 496-505)

Figure 4. Adsorption receptor identification of BUCT700. Bacteriophage BUCT700 uses Type IV fimbrial biogenesis protein PiIX as an adsorption receptor to infect *S.maltophilia* S21. (A) Compared to *S.maltophilia* S21, phage-resistant mutant S21R produced significantly impaired adsorption, whereas the complementary strain S21R + PiIX could be re-adsorbed by phage BUCT700. (B) Phage BUCT700 can rapidly adsorb *S.maltophilia* S21, whereas BUCT700 cannot adsorb phage-resistant mutant S21R. When complementation of phage-resistant mutant S21R with the PiIX gene was performed, phage BUCT700 restores adsorption to the complementary strain S21R + PiIX. (C) *S.maltophilia* S21 is susceptible to BUCT700 infection, whereas phage-resistant mutant S21R is resistant to phage infection, but the complementary strain S21R + PiIX restores phage infection to *S.maltophilia* S21 levels. Data are shown as the mean \pm SD.

8.All the sequences should be deposited in a public database with accession no.

Response: Thank you for your suggestion, we have deposited all the sequences in the NCBI databases and the accession numbers were listed in the section of Data availability.

“ The complete genome sequence of phage BUCT700 with annotations was submitted to the GenBank database under the accession number OM735686 (<https://www.ncbi.nlm.nih.gov/nuccore/OM735686>). The original sequencing data of *S.maltophilia* S21 and S21R were submitted to the GenBank database under the accession number SRX18445116 (<https://www.ncbi.nlm.nih.gov/sra/?term=SRX18445116>) and SRX18445117 (<https://www.ncbi.nlm.nih.gov/sra/?term=SRX18445117>), respectively. ” (Line 458-463)

January 8, 2023

Prof. Huahao Fan
Beijing University of Chemical Technology
No.98 Zi-Zhu-Yuan-Lu
Haidian District
Beijing 100029
China

Re: Spectrum04030-22R1 (Efficacy in *Galleria mellonella* larvae and application potential assessment of a new bacteriophage BUCT700 extensively lyse *Stenotrophomonas maltophilia*)

Dear Prof. Huahao Fan:

Your manuscript has been accepted, and I am forwarding it to the ASM Journals Department for publication. You will be notified when your proofs are ready to be viewed.

Sincerely,

Xiaoyu Tang
Editor, Microbiology Spectrum
